# PAINLESS FEDERATED LEARNING: AN INTERPLAY OF LINE-SEARCH AND EXTRAPOLATION

## ABSTRACT

The classical line search for learning rate (LR) tuning in the stochastic gradient descent (SGD) algorithm can tame the convergence slowdown due to data-sampling noise. In a federated setting, wherein the client heterogeneity introduces a slow-down to the global convergence, line search can be relevantly adapted. In this work, we show that a stochastic variant of line search tames the heterogeneity in federated optimization in addition to that due to client-local gradient noise. To this end, we introduce **Federated Stochastic Line Search (FEDSLS)** algorithm and show that it achieves deterministic rates in expectation. Specifically, FEDSLS offers *linear convergence* for strongly convex objectives *even with partial client participation*. Recently, the extrapolation of the server's LR has shown promises for improved empirical performance for federated learning. To benefit from extrapolation, we extend FEDSLS to **Federated Extrapolated Stochastic Line Search (FEDEXPSLS)** and prove its convergence. Our extensive empirical results show that the proposed methods perform at par or better than the popular federated learning algorithms across many convex and non-convex problems.

## 1 INTRODUCTION

**Federated learning.** Consider training a machine learning (ML) model $w \in \mathbb{R}^d$ on data scattered over *clients/nodes* $i \in [N]$. With limitations posed by volume, speed, governing policy, etc., on data centralization, *federated learning* (FL) is a go-to approach to train the models over client-local data. Formally, training $w \in \mathbb{R}^d$ in an FL setting is represented as $\min_{w \in \mathbb{R}^d} \left\{ f(w) := \frac{1}{N} \sum_{i=1}^{N} f(w^i) \right\}$, where $w^i \in \mathbb{R}^d$ is a local copy of $w \in \mathbb{R}^d$ on the client $i$.

A basic algorithm for the federated optimization is federated averaging (FEDAVG) (McMahan et al., 2017), where after several local SGD updates, clients synchronize at a node called *server*. FEDAVG can be described as the following:

$$w_{t,k}^i = w_{t,k-1}^i - \eta_{t,k}^i g(w_{t,k-1}^i) \text{ for } k \in [K], \text{ with } w_{t,0}^i = w_t, \tag{1}$$

$$w_{t+1} = w_t - \eta_{g_t} \Delta_t, \text{ where } \Delta_t = \frac{1}{|S_t|} \sum_{i=1}^{|S_t|} \left\{ \Delta_t^i := w_t - w_{t,K}^i \right\}, \tag{2}$$

where $w_t$ denotes the server's model after $t$ synchronization rounds, also called the *global model*. With $w_t$ communicated to clients, $w_{t,k}^i$ is the model state at client $i \in [N]$ after $k$ local gradient updates. $S_t \subseteq [N]$ is a subset of participating clients for the $t$-th round. $\Delta_t^i := w_t - w_{t,K}^i$ denotes the model update at client $i$ due to $K$ local gradient update steps, whereby, $\Delta_t$ represents the *synchronized* update to the model after $t$ rounds; $\eta_{g_t}$ is the learning rate at the server. Convergence of FEDAVG suffers from heterogeneity in clients' data distribution, their participation frequency, drift in their optimization trajectory, etc. To help mitigate these drawbacks, methods such as FEDPROX (Li et al., 2020), SCAFFOLD (Karimireddy et al., 2020), etc. were proposed. Note that the update rule (2) for $w_t$ by $\Delta_t$, often referred to as pseudo-gradient, is analogous to that of standard stochastic gradient descent (SGD) algorithm (Robbins & Monro, 1951).

**The server-side LR** $\eta_g$ naturally influences the performance of federated optimization. (Reddi et al., 2021) noted that small client LRs $\eta_{t,k}^i$ help reducing their drifts, wherein a larger server LR

$\eta_{g_t}$ can address the incurred slowdown. However, (Malinovsky et al., 2023) showed that if the clients' objectives significantly differ then larger server LR does not help convergence. Subsequently, FEDEXP (Jhunjhunwala et al., 2023) proposed using LR *extrapolation* drawing from projected convex optimization (Pierra, 1984). An extrapolated $\eta_g$ is upper-bounded by $\frac{\sum_{i \in S_t} \|\Delta_t^i\|^2}{2|S_t|(\|\Delta_t\|^2 + \epsilon)}$ and is at least 1, where $\epsilon$ is a small positive constant to avoid the cases of division by 0. Li et al. (2024) proposed FEDEXPROX by extending FEDEXP to incorporate proximal objectives on clients and showed linear convergence for strongly convex objectives under an interpolation condition across clients.

**The line search** for LR is a classical strategy proposed by Armijo (1966) that ensures guaranteed descent in function values by ensuring that $f(w_{t+1}) \leq f(w_t) - c\|\nabla f(w_t)\|^2$ for a $c > 0$ for full gradient descent. Vaswani et al. (2019) adapted it to sample-wise updates by a stochastic guarantee of $f(w_{t+1}, \xi) \leq f(w_t, \xi) - c\|g(w_t)\|^2$. They proved that under an interpolation condition generally satisfied (Zhang et al., 2016) by models such as deep neural networks, SGD with ARMIJO line search achieves the deterministic convergence rate, thereby a linear convergence for strongly convex objectives. **The deterministic rates** achieved by SGD with stochastic line search is a direct result of shielding the data sampling noise by $c\|g(w_t)\|^2$; we formally elaborate on it in Section 4. However, it is interesting to note that interpolation itself is sufficient to ensure deterministic rates, as we discuss in Section 4. Thus, it remains to investigate if ARMIJO scheme can provide expected descent without an interpolation assumption. Nevertheless, with partial participation of clients $S_t \subseteq [N]$ resulting in supplemented noise, it is imperative to translate the line search scheme to a federated setting. However, implementing line search for $\eta_g$ can not be direct because the server does not host any data sample in a standard federated setting.

Therefore, we ask if introducing line search on the clients only can tame the noise-slowdown due to both data sampling and partial client participation. Furthermore, motivated by the results of FEDEXP and FEDEXPROX, if extrapolation can further improve such an FL algorithm. Our exploration answers both these questions affirmatively. In this work, we introduce line search in federated optimization and extend it to combine with extrapolation. Our contributions are summarized below.

1. Firstly, we **strengthen and clarify the role of line search** in SGD by relaxing the assumptions of Vaswani et al. (2019) – we replace (a) sample-wise smoothness/convexity with standard population-level (expected) smoothness/convexity of the objective, and (b) interpolation with a weaker *expected sufficiently accurate function-estimates* for the stochastic functions used inside the ARMIJO condition for line search.
2. **Federated Stochastic Line Search (FEDSLS)**: We establish that the stochastic ARMIJO line-search on clients directly influences the global model update on the orchestrating server. FEDSLS provably offers deterministic rates for federated convergence even with partial participation of clients, specifically, it provides linear convergence for strongly convex objectives in this setting. Our convergence results requires an interpolation assumption only at the client-level model and escapes a requirement for this assumption at the level of samples on each client.
3. **Federated Extrapolated Stochastic Line Search (FEDEXPSLS)**: We extend FEDSLS to FEDEXPSLS that incorporates extrapolation in $\eta_g$ to harness its advantages. We prove that FEDEXPSLS provides the same convergence guarantees as FEDSLS under standard assumptions.
4. We perform extensive benchmarks to validate the empirical efficacy of the proposed algorithms. Our benchmarks prove that FEDSLS and FEDEXPSLS outperform the competitors across a variety of deep learning tasks.

Figure 1 presents the results of a toy example to motivate a reader. Similar to (Jhunjhunwala et al., 2023), we consider two clients each optimizing a distinct local objective function defined as follows:

$$F_1(\mathbf{w}) = (w_1 + w_2 - 3)^2, \quad F_2(\mathbf{w}) = (w_1 + 2w_2 - 3)^2.$$

It evidently highlights the benefits of combination of extrapolation and line search in federated learning.

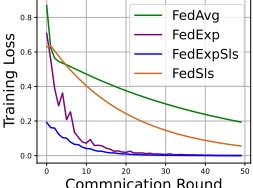

Figure 1: Efficacy of line search.

## 2 RELATED WORK

The motivations to alleviate the shortcomings of the baseline FEDAVG have led to development of a rich landscape of FL algorithms, in many cases, directly inspired by the variants of SGD.

**The data and system heterogeneity** across clients and the **associated drifts** between their optimization dynamics and the server's model's trajectory poses primary challenge for FL. For this, FEDPROX (Li et al., 2020) introduced a regularizer term: $\frac{\mu}{2}\|w^i - w\|^2$ in clients' objectives with respect to (w.r.t.) the global model making the client-local optimization proximal. Similarly, SCAFFOLD (Karimireddy et al., 2020) introduced control variates at server and clients to check the client drifts. FEDDYN (Durmus et al., 2021) proposed an additional regularization term for clients' objectives similar to FEDPROX. However, beyond a modified local objective, FEDPROX, SCAFFOLD, FEDDYN, employ the same averaging-based synchronization as given in (2) and keep the server's learning rate $\eta_g$ constant; often $\eta_g = 1$. Surely, they leave scope to tune $\eta_g$, including adapting it to $\Delta_t$ updates.

**Adaptive LR** methods such as ADAGRAD (Duchi et al., 2011), ADAM (Kingma & Ba, 2015), and YOGI (Zaheer et al., 2018), are a standard approach to improve SGD. Motivated by them, (Reddi et al., 2021) employed these schemes to update rule (2) to propose FEDADAGRAD, FEDADAM, FEDYOGI methods. Wu et al. (2023) introduced variance reduction to adaptive schemes to propose FAFED. Wang et al. (2022) introduced communication compression and error-feedback to FEDADAM.

**Beyond first order,** the second-order: FEDDANE (Li et al., 2019) and FEDNEW (Elgabli et al., 2022), and zeroth-order: (Qiu et al., 2023) model updates were also introduced to federated optimization. Furthermore, MOON (Li et al., 2021) and FEDPROTO (Tan et al., 2022) proposed model contrastive and prototype learning, respectively, in federated setting. Chatterjee et al. (2024) introduced concurrent updates on clients to harness their share-memory compute resources in a federating setting. However, none of these algorithms used any variant of line search. As we introduce the stochastic ARMIJO line search to FL, it is relevant to note other related efforts in non-federated setting.

**Variants of line search.** Classical (deterministic) line search methods include *Wolfe* conditions that include *Armijo/backtracking* (Armijo, 1966) based on sufficient decrease and *curvature/strong-Wolfe curvature* conditions (Wolfe, 1969) that add a curvature check and are standard for (L-)BFGS. *Goldstein* (Goldstein & Price, 1967)-type bracketing rules and *nonmonotone* schemes (Grippo et al., 1986; Zhang & Hager, 2004) that require the maximum/average of function values decrease have also been suggested in the deterministic regime. In stochastic regimes, two broad line search families have been explored *stochastic Armijo* tests (Vaswani et al., 2019; Paquette & Scheinberg, 2020; Cartis & Scheinberg, 2018; Berahas et al., 2021; Jin et al., 2021) that replace exact function values with mini-batch estimates and control acceptance of SGD step after line search, and *probabilistic/Bayesian* (Mahsereci & Hennig, 2017) line search methods that impose Wolfe-like conditions in expectation or with high probability. We adopt a stochastic ARMIJO-style rule embedded in SGD as the first algorithm to offer line search for federated learning.

In terms of **theoretical guarantees**, before our paper, two existing works offer linear convergence rates for strongly convex objectives: the FEDLIN algorithm (Mitra et al., 2021) and FEDEXPROX of (Li et al., 2024). FEDLIN achieves linear ergodic convergence – convergence of function of averaged model over iterates – for smooth and strongly convex objectives with full gradient updates and full client participation. In the stochastic setting, FEDLIN maintains a standard sublinear convergence even for strongly convex objectives. By contrast, our method provides a linear convergence even with the stochastic gradient updates and partial client participation. The experimental performance of FEDLIN is not known beyond a basic linear regression on a small dataset. As mentioned before, FEDEXPROX offers deterministic rates similar to us. Our experimental results in Section 5 show that FEDEXPSLS outperforms FEDEXPROX in many cases.

## 3  ALGORITHM AND ASSUMPTIONS

**The interface** for the FEDSLS and FEDEXPSLS algorithms is given as a pseudo-code in Algorithm 1. We refer to Vaswani et al. (2019)'s algorithm as SGD-ARMIJO. The complete SGD-ARMIJO (Algorithm 2), FEDSLS (Algorithm 4), and FEDEXPSLS (Algorithm 5) are given in Appendixes A and B. Essentially, each client conducts local gradient update using SGD-ARMIJO method, while the server opts to extrapolate its LR. We now state some standard assumptions:

**Assumption 1** (Smoothness). The functions $f_i$ are $L$-smooth, i.e., for all $x, y \in \mathbb{R}^d$, it holds that $f_i(y) \leq f_i(x) + \nabla f_i(x)^\top (y - x) + \frac{L}{2}\|y - x\|^2$. It is straightforward to prove that $f$ as a sum of $L$-smooth functions is also $L$-smooth.

**Assumption 2** (Convexity). When needed, we specify that the functions $f_i$ are convex, i.e., for all $x, y \in \mathbb{R}^d$, $f_i(y) \geq f_i(x) + \nabla f_i(x)^\top (y - x)$. Therein, $f$ is also convex.

**Algorithm 1** A framework for FEDSLS and FEDEXPSLS methods.

---
1: initialize $w_0$
2: **for** each round $t = 0, 1, \ldots, T-1$ **do**
3:  $\quad \mathcal{S}_t \leftarrow$ (random set of $S$ clients); Server sends $w_t$ to clients $i \in \mathcal{S}_t$ in parallel
4:  $\quad$ **for** each client $i \in \mathcal{S}_t$ **do**
5:  $\quad\quad$ **for** $k = 1, 2, \ldots, K$ **do**
6:  $\quad\quad\quad w_{t,k}^i \leftarrow \text{SGD}-\text{ARMIJO}(w_{t,k-1}^i)$
7:  $\quad\quad$ **end for**
8:  $\quad\quad \Delta_t^i \leftarrow w_t - w_{t,K}^i$
9:  $\quad$ **end for**
10: $\quad \Delta_t = \frac{1}{S} \sum_{i \in \mathcal{S}_t} \Delta_t^i; \eta_{g_t} \leftarrow \max\left\{1, \frac{\sum_{i \in \mathcal{S}_t} \|\Delta_t^i\|^2}{2|\mathcal{S}_t|(\|\Delta_t\|^2 + \epsilon)}\right\}$ if FEDEXPSLS else $\eta_g$ if FEDSLS;
11: $\quad w_{t+1} \leftarrow w_t - \eta_{g_t} \Delta_t$
12: **end for**
13: **return** $w_T$

---

**Assumption 3** (Strong- Convexity). When needed, we specify that the functions $f_i$ are $\mu-$ strongly convex, i.e., for all $x, y \in \mathbb{R}^d$, it holds that $f_i(y) \geq f_i(x) + \nabla f_i(x)^\top (y-x) + \frac{\mu}{2}\|y-x\|^2$. Therein, $f$ is also $\mu-$ strongly convex.

We also lay out the following additional assumptions which we use in the discussions but are not assumed for our theoretical results:

**Assumption 4** (Bounded Variance). We assume that the variance of $g_{t,k}^i(w)$ is bounded by a constant $\sigma^2$, given as $\mathbb{E}[\|g_{t,k}^i(w) - \nabla f_i(w)\|^2] \leq \sigma^2$.

**Assumption 5** (Bounded Gradient dissimilarity). The norm of the clients' gradient averaged across all clients for all $w \in \mathbb{R}^d$ is bounded as $\frac{1}{N} \sum_{i=1}^N \|\nabla f_i(w)\|^2 \leq G^2 + B^2 \|\nabla f(w)\|^2$, for $G \geq 0$, $B \geq 1$.

If $f_i$ are convex, then the bound can be relaxed to $\frac{1}{N} \sum_{i=1}^N \|\nabla f_i(w)\|^2 \leq G^2 + 2LB^2(f(w) - f(w^*))$.

## 4 CONVERGENCE RESULTS

**Definition 1** (Armijo Condition). *For the $k$-th step in the $t$-th communication round, the Armijo condition for the local objective functions $f_i$ at a sample $\xi_k$ with a constant $c > 0$ is given by*

$$f_i(w_{t,k}^i, \xi_k) - f_i(w_{t,k-1}^i, \xi_k) \leq -c\eta_{t,k}^i \|g_i(w_{t,k-1}^i)\|^2. \tag{3}$$

### 4.1 DETERMINISTIC RATES FOR SGD

Here we discuss how ARMIJO condition mitigates the effect of the bias term in convergence of SGD and retrieves deterministic GD rates in expectation. For brevity, we drop the subscript $t$ and superscript $i$ here as we are looking at the SGD updates at a single client for local rounds.

Denote the loss function for $i$-th client performing SGD update by $f_i(w) := \frac{1}{M_i} \sum_{m=1}^{M_i} f_i(w, \xi^m)$, where $\xi^m$ denotes the $m$-th sample and $M_i$ is the total number of samples for the client $i$. The stochastic gradient $g_i(w) := \nabla f_i(w, \xi)$ is the unbiased estimator of the full gradient $\mathbb{E}[g_i(w)] = \nabla f_i(w)$. We first give a few definitions.

**Definition 2** (Sample-wise Interpolation). *For a sum of functions problem, if there exists a $w^* \in \mathbb{R}^d$ such that $f_i(w^*, \xi^m) = \inf_w f_i(w, \xi^m)$ for all $m = 1, 2, \ldots, M_i$, then interpolation holds, i.e., $g_i(w^*) = \nabla f_i(w^*, \xi^m) = 0$*

We now define the notion of *expected sufficiently accurate stochastic estimates* for a single local solver to analyze SGD by reformulating the probabilistically sufficiently accurate function estimate definition in Paquette & Scheinberg (2020).

**Definition 3** ($\kappa_f^i$-accurate function). *For $c > 0$ as in definition 1 and for $0 < \kappa_f^i < \frac{c}{2\eta_{l_{\max}}}$, the stochastic function estimates $f_i(w_{k-1}^i, \xi_k)$ and $f_i(w_k^i, \xi_k)$, at the sample $\xi_k$ drawn independently*

*at random at step $k$, of the true functions $f_i(w_{k-1}^i)$ and $f_i(w_k^i)$, respectively, are $\kappa_f^i$- accurate in expectation with respect to the current iterate $w_{k-1}^i$, step-size $\eta_k^i$, and the stochastic gradient $g_i(w_{k-1}^i) := \nabla f_i(w_{k-1}^i, \xi_k)$ for a sample $\xi_k$ if it holds that*

$$\mathbb{E}\left[|f_i(w_{k-1}^i, \xi_k) - f_i(w_{k-1}^i)| \,\big|\, \mathcal{F}_{k-1}\right] \leq \kappa_f^i \,\mathbb{E}\left[(\eta_k^i)^2 \|g_i(w_{k-1}^i)\|^2 \big| \mathcal{F}_{k-1}\right],$$

$$\mathbb{E}\left[|f_i(w_k^i, \xi_k) - f_i(w_k^i)| \,\big|\, \mathcal{F}_{k-1}\right] \leq \kappa_f^i \,\mathbb{E}\left[(\eta_k^i)^2 \|g_i(w_{k-1}^i)\|^2 \big| \mathcal{F}_{k-1}\right],$$

*where $\mathcal{F}_{k-1}$ is the filtration that accounts for all the randomness due to stochastic function and gradient estimates up to step $(k-1)$.*

Define $\kappa_f := \max\limits_{i \in [N]} \kappa_f^i$. Thus, $f_i \,\forall i \in [N]$ is $\kappa_f$-accurate in expectation.

**Remark 1.** For linear least-squares loss function, the interpolation condition trivially satisfies the assumption that the function estimates are sufficiently accurate in expectation, since LHS=RHS= 0 as $f_i(w^*, \xi) = 0 = f(w^*)$ for any $\xi$ and $g_i(w^*) = 0$. Thus, in this particular case, interpolation is a stronger condition than the assumption that function estimates are sufficiently accurate in expectation.

For $\eta_l < \frac{1}{2L}$, for smooth and convex objectives, classical SGD iterates satisfy (see, Appendix A.1)

$$\mathbb{E}\left[f_i(\bar{w}_k) - \inf f_i\right] \leq \frac{1}{2\eta_l K(1 - 2\eta_l L)} \|w_0 - w^*\|^2 + \underbrace{\frac{\eta_l \sigma_f^*}{(1 - 2\eta_l L)}}_{\text{bias term}}, \tag{4}$$

where $\sigma_{f_i}^* := \inf_{w^* \in \arg\min f_i} \mathbb{E}\|g_i(w^*) - \nabla f_i(w^*)\|^2$ and $\inf f$ denotes a lower bound for $f(w), \forall w \in \mathbb{R}^d$. It is easy to see that the performance of SGD slows down compared to GD due to the presence of the bias term which depends on variance of the gradient noise. We now present a result of SGD with ARMIJO line-search, which depicts how ARMIJO condition allows overcome this bias *without* sample-wise interpolation.

**Theorem 1.** *Let the objective function for the $i$-th device $f_i$ be $L$-smooth and convex, the function estimates in ARMIJO line-search are $\kappa_f$ sufficiently accurate in expectation. For $c > \frac{1}{2} + \kappa_f \eta_{l_{\max}}$, SGD with ARMIJO Line search (3) achieves the convergence rate of deterministic gradient descent in expectation as*

$$\mathbb{E}\left[f_i(\bar{w}_k) - \inf f_i\right] \leq \frac{\tilde{c}}{(2\tilde{c} - 1)\eta_{l_{\max}} K} \|w_0 - w^*\|^2 \tag{5}$$

*where $\tilde{c} := c - 2\kappa_f \eta_{l_{\max}}$ and $\bar{w}_k = \frac{1}{K} \sum_{k=1}^K w_{k-1}$.*

Note that when $c > \frac{1}{2}$, the rate given by the bound in (5) is satisfied unlike classical SGD, where the LR is tuned manually, which is similar to the results given by Vaswani et al. (2019).

Comparing (4) and (5), we can see that Armijo condition aids in mitigating the effect of gradient noise. These results highlight the explicit benefit of the ARMIJO condition in stochastic settings. We defer the complete proof to Appendix A.1, where we also discuss the cases for other analytical classes of functions. Motivated by this insight, we use ARMIJO line search in Federated Learning to mitigate the effect of client drift and gradient noise.

## 4.2 TOWARDS DETERMINISTIC RATES IN FEDERATED LEARNING

We now discuss the impact of SGD-ARMIJO, when implemented as client-local solver. The SGD updates at a client $i$ after $K$ local steps is written as $w_{t,K}^i = w_t - \sum_{k=1}^K \eta_{t,k}^i \, g_i(w_{t,k-1}^i)$. The theoretical results correspond to the partial participation of clients. For brevity, we $\sum_k$ to denote $\sum_{k=1}^K$, $\sum_i$ to denote $\sum_{i=1}^N$, and $\sum_{i \in \mathcal{S}_t}$ denotes summation over $i \in \mathcal{S}_t$. We define a filtration $\mathcal{F}_t$ that contains all the randomness up to the evaluation of the global update $w_t$. We now state the assumption of sufficient accurate function estimates in expectation for the federated setting, a natural extension of Definition 3.

**Assumption 6.** For some $0 < \kappa_f^i < \dfrac{c}{2\eta_{l_{\max}}}$, the stochastic function estimates $f_i(w_{t,k-1}^i, \xi_k^i)$ and $f_i(w_{t,k}^i, \xi_k^i)$, at the sample $\xi_k^i$ drawn independently at random at local round $k$ in $t$-th global round, of

the true functions $f_i(w_{t,k-1}^i)$ and $f_i(w_{t,k}^i)$, respectively are $\kappa_f^i$- accurate in expectation with respect to the current iterate $w_{t,k-1}^i$, step-size $\eta_{t,k}^i$ and $g_i(w_{t,k-1}^i) := \nabla f_i(w_{t,k-1}^i, \xi_k^i)$ for a sample $\xi_k^i$, i.e.,

$$\mathbb{E}\left[|f_i(w_{t,k-1}^i, \xi_k^i) - f_i(w_{t,k-1}^i)| \,\big|\, \mathcal{F}_{t,k-1}^i\right] \leq \kappa_f^i \, \mathbb{E}\left[(\eta_{t,k}^i)^2 \|g_i(w_{t,k-1}^i)\|^2 \big| \mathcal{F}_{t,k-1}^i\right]$$

$$\mathbb{E}\left[|f_i(w_{t,k}^i, \xi_k^i) - f_i(w_{t,k}^i)| \,\big|\, \mathcal{F}_{t,k-1}^i\right] \leq \kappa_f^i \, \mathbb{E}\left[(\eta_{t,k}^i)^2 \|g_i(w_{t,k-1}^i)\|^2 \big| \mathcal{F}_{t,k-1}^i\right]$$

where $\mathcal{F}_{t,k-1}^i$ denote the $\sigma$-algebra containing $\mathcal{F}_t$ and all local randomness of client $i$ up to step $k-1$. Define $\kappa_f := \max_{1,...,N} \kappa_f^i$, thus for some $0 < \kappa_f < \frac{c}{2\eta_{l_{\max}}}$, the assumption of $\kappa_f$-accurate function in expectation holds.

We now present Lemma 1 that highlights that client-local SGD-ARMIJO alleviates the requirement for heterogeneity bound in federated setting.

**Lemma 1.** *Under assumption 6, there exists $c' := (c - 2\kappa_f \eta_{l_{\max}}) > 0$, equivalently, $\kappa_f < \frac{c}{2\eta_{l_{\max}}}$, such that* ARMIJO *line-search (3) yields*

$$\sum_{k,i} \mathbb{E}\left[\|\nabla f_i(w_{t,k-1}^i)\|^2\right] \leq \max\left\{\frac{L}{2(1-c)}, \frac{1}{\eta_{l_{\max}}}\right\} \frac{1}{c'} \left(f(w_t) - \mathbb{E}\left[\sum_{i \in \mathcal{S}_t} \frac{1}{S} f_i(w_{t,K}^i)\right]\right). \quad (6)$$

The proof is deferred to Appendix E.

**ARMIJO line search vs. bounded heterogeneity** The standard bounded heterogeneity assumption 5 for the iterate $w_{t,k-1}^i$ at $t, k$ step can be given as $\frac{1}{N} \sum_{i,k} \mathbb{E}\left[\|\nabla f_i(w_{t,k-1}^i)\|^2\right] \leq G^2 + B^2 \sum_k \mathbb{E}\left[\|\nabla f(w_{t,k-1}^i)\|^2\right]$, for $G \geq 0$, $B \geq 1$. Comparing it to (6), we can see that ARMIJO line search provides another upper bound for the same quantity and thus, we can prove the results without needing assumption 5. However, it is difficult to resolve the term $\sum_{i \in \mathcal{S}_t} \frac{1}{S} f_i(w_{t,K}^i)$ to the global objective function at some known argument. To resolve this, we need to adapt the client-wise interpolation assumption for our results.

**Remark 2.** In the special case, $f_i \equiv f$ for all $i \in [N]$ (i.e., $G = 0, B \geq 1$), a case stronger than i.i.d. data , then due to the convex (or strongly convex) nature of functions, Jensen's inequality enables carrying the clients' objective's descents to the global objective when the global learning rate $\eta_{g_t} \leq 1$. Thus, in that case for convex objectives, the descent in the global function comes for free. However, descent can't be guaranteed for non-convex objectives.

**Assumption 7** (Client-wise interpolation). There exists $w^* \in \mathbb{R}^d$ such that $\nabla f_i(w^*) = 0$ for all $i \in \{1, 2, \ldots, N\}$.

With assumptions 6 and 7, ARMIJO line search enables a simplified convergence analysis of the proposed algorithms FEDSLS and FEDEXPSLS. It allows for control of the client drift using an objective gap between the current global iterate and the averaged local objectives after $k$ local ARMIJO line search calls– $f(w_t) - \frac{1}{S} \sum_{i \in \mathcal{S}_t} f_i(w_{t,K}^i)$, rather than relying on auxiliary bias terms that obtained using bounded heterogeneity and bounded variance of the clients' gradients. This structural advantage is the reason we are able to achieve convergence rates comparable to those in the deterministic full-participation setting, despite the presence of partial client participation and stochastic gradient updates. However, we need the interpolation regime (client-wise) to translate the average of local objectives $\frac{1}{S} \sum_{i \in \mathcal{S}_t} f_i(w_{t,k}^i)$ to the global objective evaluated at some value. While under interpolation, one can relate $f_i(w_t) - \frac{1}{s} \sum_{i \in \mathcal{S}_t} f_i(w_{t,K}^i)$ to the optimality gap $f(w_t) - f(w^*)$.

Note that Li et al. (2024) also achieved linear rates for strongly convex objectives. At the core of their approach lies the proximal term in the client objectives. An exact solution of the proximal problem largely provides the foundation for mitigating the bias term in convergence error upper-bound that we used the ARMIJO condition for. They extended their results in (Anyszka et al., 2024) to show that FEDEXPROX achieves linear convergence for Polyak-Lojasiewicz objectives.

## 4.3   CONVERGENCE OF FEDSLS

We now describe the convergence results of FEDSLS for convex, strongly convex and non-convex classes of objective functions.

**Theorem 2** ($f_i$ are convex). *Let the functions $f_i$ satisfy the assumptions 1, 2, 6 and 7. For a constant global learning rate $\eta_{g_t} = \eta_g$ and client learning rate $\eta_{l_{\max}} < \frac{2c - \eta_g}{(KL + 4\kappa_f)}$, FEDSLS achieves the convergence rate for average of iterates as*

$$\mathbb{E}[f(\bar{w}_t) - f(w^*)] \leq \frac{c'}{(2c' - \eta_g - KL\eta_{l_{\max}})\eta_g \eta_{l_{\max}} KT} \|w_0 - w^*\|^2 \qquad (7)$$

*where $\bar{w}_t = \frac{1}{T}\sum_{t=0}^{T-1} w_t$ and $c' := c - 2\kappa_f \eta_{l_{\max}}$, such that $c' > 0$.*

The proof of Theorem 2 is included in Appendix C in the supplementary. Theorem 2 shows a sublinear convergence for convex problems.

**Theorem 3** ($f_i$ are strongly convex). *Let the functions $f_i$ satisfy the assumptions 1, 3, 6 and 7. For a constant global learning rate $\eta_{g_t} = \eta_g$ such that $\eta_g \leq \frac{2}{\eta_{l_{\max}}\mu K}$, client learning rate $\eta_{l_{\max}} < \frac{2c - \eta_g}{KL + 4\kappa_f}$, FEDSLS algorithm satisfies*

$$\mathbb{E}\|w_T - w^*\|^2 \leq \left(1 - \frac{\eta_g \eta_{l_{\max}}\mu K}{2}\right)^T \|w_0 - w^*\|^2.$$

The proof of Theorem 3 is included in the supplementary in Appendix C. Theorem 3 shows a *linear convergence* for strongly-convex problems.

**Theorem 4** ($f_i$ are non-convex). *Let functions $f_i$ satisfy the assumptions 1, 6 and 7. For $\eta_{l_{max}} \geq \frac{\frac{8c'}{2T-1}}{\eta_g^2 LK + \sqrt{(\eta_g^2 LK)^2 + \eta_g L^2 K^2 \frac{16c'}{2T-1}}}$, FEDSLS achieves the convergence rate*

$$\min_{t=0,\ldots,\,T-1} \mathbb{E}[\|\nabla f(w_t)\|^2] \leq \frac{2L(\eta_{l_{\max}}LK + \eta_g)}{c'}\mathbb{E}[f(w_0) - f(w^*)],$$

*where $c' := c - 2\kappa_f \eta_{l_{\max}} > 0$.*

The details are included in Appendix C in the supplementary. Theorem 4 shows a sub-linear convergence for non-convex problems.

## 4.4 CONVERGENCE OF FEDEXPSLS

We now describe the convergence results of FEDEXPSLS for convex, strongly convex and non-convex classes of objective functions.

**Theorem 5** ($f_i$ are convex). *Suppose a function $f_i$ satisfy assumption 1 2, 6 and 7. For global learning rate $\eta_{g_t}$ as computed in FEDEXPSLS constrained to lie in $[1, \eta_{g_{\max}}]$, client learning rate $\eta_{l_{\max}} < \frac{2c' - 1}{KL + 4\kappa_f}$, FEDEXPSLS achieves the convergence rate for average of iterates as*

$$\mathbb{E}\left[f(\bar{w}_t) - f(w^*)\right] \leq \frac{c'}{(2c' - \eta_{l_{\max}}KL - 1)\eta_{l_{\max}}\eta_{g_{\max}}KT}\|w_0 - w^*\|^2,$$

*where $\bar{w}_t = \frac{1}{T}\sum_{t=1}^{T} w_t$ and $c'$.*

The proof of Theorem 5 is included in Appendix C in the supplementary. Theorem 5 shows a sublinear convergence for convex problems.

**Theorem 6** ($f_i$ are strongly convex). *Let the functions $f_i$ satisfy assumption 1, 3, 6 and 7. For a global learning rate $\eta_{g_t}$ computed in FEDEXPSLS constrained to lie in $[1, \eta_{g_{\max}}]$ such that $\eta_{g_{\max}} \leq \frac{2}{\eta_{l_{\max}}\mu K}$, client learning rate $\eta_{l_{\max}} < \frac{2c - 1}{KL + 4\kappa_f}$, the last update of FEDEXPSLS satisfies*

$$\mathbb{E}\|w_{T+1} - w^*\|^2 \leq \left(1 - \frac{\eta_g \eta_{l_{\max}}\mu K}{2}\right)^{T+1} \|w_0 - w^*\|^2.$$

The proof of Theorem 6 is included in the supplementary in Appendix C in the supplementary. Theorem 6 shows a *linear convergence* for strongly-convex problems.

**Theorem 7** ($f_i$ are non-convex). *Let the functions $f_i$ satisfy assumption 1, 6 and 7. For a global learning rate $\eta_{g_t}$ computed in* FEDEXPSLS *constrained to lie in $[1, \eta_{g_{\max}}]$ and local learning rate bound $\eta_{l_{max}} \geq \frac{\frac{8c'}{2T-1}}{\eta_{g_{\max}} LK + \sqrt{(\eta_{g_{\max}} LK)^2 + \eta_{g_{\max}} L^2 K^2 \frac{16c'}{2T-1}}}$,* FEDSLS *achieves the convergence rate*

$$\min_{t=0,\ldots,\,T-1} \mathbb{E}[\|\nabla f(w_t)\|^2] \leq \frac{2L(\eta_{l_{\max}} LK + 1)}{c'} \mathbb{E}[f(w_0) - f(w^*)],$$

*where $c' := c - 2\kappa_f \eta_{l_{\max}} > 0$.*

The details are included in Appendix C in the supplementary. Theorem 7 shows a sub-linear convergence for non-convex problems.

## 5 EXPERIMENTS AND NUMERICAL RESULTS

In this section, we conduct comprehensive evaluation of the proposed federated optimizers by experimentally comparing their performance against established federated algorithms: FEDAVG, FEDEXP, FEDEXPROX. We also include FEDADAM in the benchmarks for language model and in a high heterogeneity case. The objective is to demonstrate that FEDEXPSLS leads to faster convergence and improved stability during training in communication rounds.

**Datasets and Architecture:** We evaluated the proposed algorithms on a diverse set of benchmarks that cover image classification and text prediction tasks. Our experiments involved four combination of datasets (Caldas et al., 2018) and models: (a) **CIFAR-10** with ResNet-18, (b) **CIFAR-100** with ResNet-18, (c) **FEMNIST** with Multi-Class Logistic Regression, and (d) **SHAKESPEARE** with Long Short-Term Memory (LSTM).

**Experimental Setup** For training across different algorithms, we distributed CIFAR-10 and CIFAR-100 over 100 clients as in (Jhunjhunwala et al., 2023). The number of clients for **FEMNIST** and **SHAKESPEARE** are selected as in (Caldas et al., 2018). In each training round, we uniformly sample 20 clients without replacement within a round, but with replacement across rounds. We compute mini-batch gradients on each client using a fixed batch size of 50. The number of local epochs is fixed at $K = 20$ for all experiments. To introduce heterogeneity in the data distribution across clients, we employ a Dirichlet distribution with a concentration parameter $\alpha = 0.3$ (Caldas et al., 2018), which is standard in the existing experimental benchmarks (Karimireddy et al., 2020). The training loss is calculated as the average of the losses reported by the participating clients in each round, aggregated over 5 runs using different random seeds. All experiments were performed on NVIDIA A6000 GPUs with 48 GB onboard memory. Wherever required, we performed grid search for hyperparameter tuning.

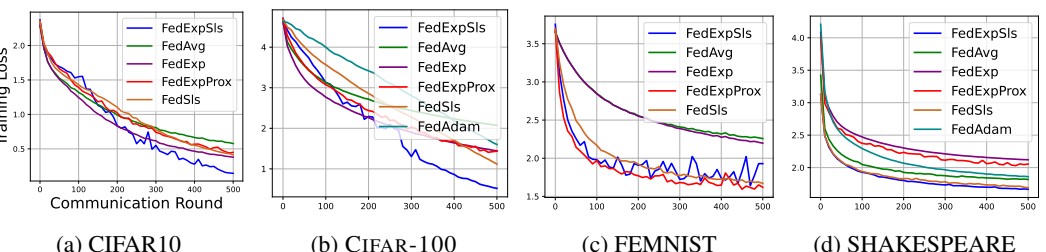

(a) CIFAR10      (b) CIFAR-100      (c) FEMNIST      (d) SHAKESPEARE

Figure 2: Training loss v/s Communication Rounds

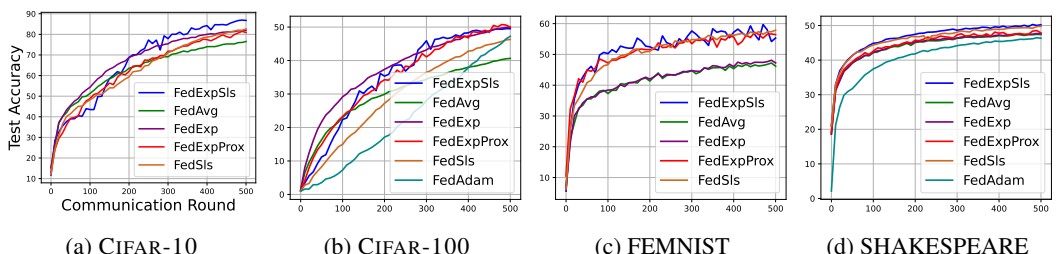

| (a) CIFAR-10 | (b) CIFAR-100 | (c) FEMNIST | (d) SHAKESPEARE |

Figure 3: Test Accuracy v/s Communication Rounds

Table 1: Comparison of Training Loss

| Method | CIFAR-10 | CIFAR-100 | FEMNIST | SHAKESPEARE |
|---|---|---|---|---|
| FEDAVG | $0.57 \pm 0.01$ | $2.07 \pm 0.02$ | $2.25 \pm 0.00$ | $1.81 \pm 0.01$ |
| FedExp | $0.38 \pm 0.01$ | $1.44 \pm 0.02$ | $2.19 \pm 0.00$ | $2.12 \pm 0.02$ |
| FedExpSLS | $\mathbf{0.13 \pm 0.01}$ | $\mathbf{0.5 \pm 0.05}$ | $1.6 \pm 0.01$ | $\mathbf{1.66 \pm 0.03}$ |
| FedExpProx | $0.43 \pm 0.06$ | $1.35 \pm 0.13$ | $\mathbf{1.57 \pm 0.001}$ | $2.03 \pm 0.03$ |
| FEDSLS | $0.41 \pm 0.03$ | $1.12 \pm 0.05$ | $1.65 \pm 0.002$ | $1.69 \pm 0.001$ |

Table 2: Comparison of Test Accuracy

| Method | CIFAR-10 | CIFAR-100 | FEMNIST | SHAKESPEARE |
|---|---|---|---|---|
| FEDAVG | $76.8 \pm 0.54$ | $40.94 \pm 0.38$ | $47.90 \pm 0.09$ | $47.67 \pm 0.01$ |
| FEDEXP | $82.09 \pm 0.56$ | $50.24 \pm 0.34$ | $48.68 \pm 0.21$ | $47.81 \pm 0.23$ |
| FEDEXPSLS | $\mathbf{87.29 \pm 0.6}$ | $50.23 \pm 3.37$ | $\mathbf{60.92 \pm 0.42}$ | $\mathbf{50.37 \pm 0.03}$ |
| FEDEXPROX | $81.79 \pm 1.48$ | $\mathbf{52.13 \pm 2.18}$ | $58.13 \pm 3.06$ | $48.56 \pm 0.12$ |
| FEDSLS | $82.75 \pm 0.81$ | $46.53 \pm 0.6$ | $58.47 \pm 0.12$ | $49.94 \pm 0.35$ |

Our code is available at `https://anonymous.4open.science/r/FederatedLineSearch-B663/README.md`.

**Analysis of Results** The results of the experiments are shown as comparative training loss and test accuracy in Figures 2 and 3. We also present the numerical results with standard deviation in Tables 1 and 2. Across all experiments, FEDEXPSLS consistently outperforms other algorithms. As an exception to this trend, for the CIFAR-100 dataset, FEDEXPROX marginally does better in terms of test accuracy. The high class count (100 classes) in CIFAR-100 introduces greater heterogeneity, which favors the performance of FEDEXPROX. We evaluated FEDADAM for CIFAR-100 and SHAKESPEARE datasets that involve higher heterogeneity and language models, respectively. However, in both cases it overwhelmingly underperforms. We also counted the number of retries in both FEDSLS and FEDEXPSLS to check the overhead for descent guarantees. However, in no case we found the numbers higher than 2 in any round of training, which promises a very light overhead. Our experimental results comprehensively back the algorithmic efficacy of our algorithms.

## 6 CONCLUSION AND DISCUSSION

In this paper we introduced two new federated learning algorithms. The algorithm FEDSLS provides convergence rates similar to deterministic gradients even with partial participation of clients. Our work uncovers that a line search scheme for the client-local stochastic gradient updates can tame the effect of heterogeneity thus removing the requirement for an explicit bound on the same. Practically, the line search addresses the slow down due to partial client participation, in addition to the data and system-induced heterogeneity. The algorithm FEDEXPSLS empirically outperforms the state-of-the-art methods across deep learning tasks. Our approach motivates exploring and extending line search to future federated optimization algorithms.

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

# APPENDIX

## A   ARMIJO LINE SEARCH ALGORITHM

Algorithm 2 gives the pseudo-code for SGD with ARMIJO line-search implemented via the call SGD-ARMIJO$\big(w_{k-1}^i, \eta, \eta_{l,\max}, \delta, b, k, \texttt{opt}\big)$ for $k = 1, 2, \ldots, K$, where $w_{k-1}^i$ is the state of the model at $(k-1)$-th local round on a client with loss function $f_i$; $\eta_{l_{\max}}$ is an upper bound on the step size $\eta$, $\delta$ is the scaling/reset factor, $b$ is the minbatch size , and $\texttt{opt}$ is the reset policy. The line-search scheme is stochastic because the Armijo condition 3 is evaluated on a minibatch (potentially of size 1) to compute function $f_i$ and its gradient. We include the SGD algorithm with ARMIJO line-search 2 from the point of view of a client (Vaswani et al., 2019).

---

**Algorithm 2** SGD$-$ARMIJO$(w_{k-1}^i, \eta, \eta_{l_{\max}}, \delta, b, k, \texttt{opt})$.

---

**Input**: $\eta_{l_{\max}}$, $b$, $c$ the ARMIJO parameter, $\beta$ the backtracking factor, $\delta > 1$, and $\texttt{opt}$

**Output**: $w_k^i$

1: $b_k \leftarrow$ sample mini-batch of size $b$

2: $\eta \leftarrow \text{RESET}/\beta$

3: **repeat**

4:      $\eta \leftarrow \beta \cdot \eta$

5:      $\tilde{w}_k^i \leftarrow w_{k-1}^i - \eta \nabla f_i(w_{k-1}^i, b_k)$

6: **until** $f_i(\tilde{w}_k^i, b_k) \leq f_i(w_{k-1}^i, b_k) - c \cdot \eta \|\nabla f_i(w_{k-1}^i, b_k)\|^2$                    ▷ **Armijo Line Search**

7: $w_k^i \leftarrow \tilde{w}_k^i$

8: **return** $w_k^i$

---

The RESET method, given in Algorithm 3, heuristically resets $\eta$ based on the handle $\texttt{opt}$ at every gradient update step. Taking $\eta_{\max} = \eta_{k-1}$ could be one strategy where we start dampening the step-size from the last achieved state. However, it can increase the backtracking. This method can

implement various heuristics that appeared in the literature: (Nocedal & Wright, 1999).Chapter 3. The heuristic line search is an active area of research with new developments such as a new variant of Goldstein Line search by Neumaier & Kimiaei (2024).

---

**Algorithm 3** RESET

---

**Input**: $\eta$, $\eta_{l_{\max}}$, $b$, $k$, $\delta > 1$, and `opt`
**Output**: $w_k^i$
  1: **if** $k = 1$ **then**
  2:     **return** $\eta_{l_{\max}}$
  3: **else if** `opt` $= 0$ **then**
  4:     $\eta \leftarrow \eta$
  5: **else if** `opt` $= 1$ **then**
  6:     $\eta \leftarrow \eta_{l_{\max}}$
  7: **else if** `opt` $= 2$ **then**
  8:     $\eta \leftarrow \eta \cdot \delta^{\frac{b}{n}}$
  9: **end if**
10: **return** $\eta$

---

### A.1 DISCUSSION ON DETERMINISTIC LEARNING RATE

The convergence rate of SGD is slower in comparison to GD due to the effect of the variance of stochastic gradients. This leads to SGD requiring a larger number of epochs to achieve the same error tolerance in comparison to GD. In the following section, we explore how line-search using ARMIJO rule mitigates this effect of variance and allows us to achieve deterministic rates for SGD in expectation.

Let us consider $f_i$ to be the loss function for the $i$-th device performing an SGD update. In stochastic setting, $f_i$ is defined as $f_i(w) := \frac{1}{M} \sum_{m=1}^{M} f(w, \xi^m)$, where $\xi^m$ denotes the $m$-th sample and $M$ is the total number of samples for the device. The stochastic gradient $g_i(w) := \nabla f_i(w, \xi)$ is the **unbiased estimator** of the full gradient $\mathbb{E}[g_i(w)] = \nabla f_i(w)$. In this section, we drop the subscript $t$ and superscript $i$ from the model updates at a client $w_{t,k}^i$ and write it as $w_k^i$ instead, as we are looking at the SGD updates at a single client for local rounds.

We first state a few prerequisites for our discussion.

**Definition 4** (Interpolation). *For a sum of functions problem, if there exists a $w^* \in \mathbb{R}^d$ such that $f_i(w^*, \xi^m) = \inf_w f_i(w, \xi^m)$ for all $m = 1, 2, \ldots, M$, then interpolation holds.*

**Lemma 2** (Variance transfer). *Define $\sigma_{f_i}^* := \inf_{w^* \in \arg\min f_i} \mathbb{E}\|g_i(w^*) - \nabla f_i(w^*)\|^2$. If each $f(w, \xi^m)$ is convex and $L$-smooth, then for every $w \in \mathbb{R}^d$, we have*

$$\mathbb{E}\|g_i(w)\|^2 \leq 4L(f_i(w) - \inf f_i) + 2\sigma_{f_i}^*$$

The SGD update can be written as: $w_k^i = w_{k-1}^i - \eta_k^i g_i(w_{k-1}^i)$, where $\eta_k^i$ is the learning rate, $w_{k-1}^i$ is the SGD update at $(k-1)$-th step. The proof of classical SGD for convex objectives $f_i$ with a fixed learning rate $\eta_k^i = \eta_l$ using first-order convexity and lemma 2 gives the following bound:

$$\mathbb{E}\left[\|w_k^i - w^*\|^2 | w_{k-1}\right] \leq \|w_{k-1}^i - w^*\|^2 + 2\eta_l \langle \nabla f_i(w_{k-1}^i), w_{k-1}^i - w^* \rangle + \mathbb{E}\left[\|g_i(w_{k-1}^i)\|^2 | w_{k-1}\right]$$

$$\leq \|w_{k-1}^i - w^*\|^2 + 2\eta_l(2\eta_l L - 1)(f_i(w_{k-1}^i) - \inf f_i) + 2\eta_l^2 \sigma_{f_i}^*$$

$$\leq \|w_{k-1}^i - w^*\|^2 - 2\eta_l(1 - 2\eta_l L)(f_i(w_{k-1}^i) - \inf f_i) + 2\eta_l^2 \sigma_{f_i}^* \qquad (8)$$

for $0 < \eta_l < \frac{1}{2L}$. Averaging on both sides for $k = 1, 2, \ldots, K$, rearranging the terms and substituting $\bar{w}_k^i = \frac{1}{K} \sum_{k=1}^{K} w_{k-1}^i$ after using Jensen's inequality, we obtain

$$\mathbb{E}\left[f_i(\bar{w}_k^i) - \inf f_i\right] \leq \frac{1}{2\eta_l(1 - 2\eta_l L)K} \|w_0^i - w^*\|^2 + \underbrace{\frac{\eta_l}{(1 - 2\eta_l L)}}_{bias \; term} \sigma_{f_i}^* \qquad (9)$$

The bias term in 9 represents the slowdown in convergence compared to deterministic GD. This term can be subsumed under the interpolation condition 4, when $\sigma^*_{f_i} = 0$, the rate obtained in Equation 9 is that of deterministic GD. (Vaswani et al., 2019) et al. retrieved the deterministic GD rates for SGD using ARMIJO line-search in expectation under the interpolation condition. This does not actually reflect the benefit of using ARMIJO line-search.

We now discuss how ARMIJO condition subsumes the bias term in SGD and retrieves deterministic GD rates in expectation. We begin with the assumption of expected sufficiently accurate stochastic estimates for a single local solver to analyze SGD. Note that $\mathcal{F}_{k-1}$ is the filtration that accounts for all the randomness due to stochastic function and gradient estimates up to step $(k-1)$.

**Assumption 8.** Define $\kappa_f := \max_{i \in [N]} \kappa^i_f$. We assume that for some $0 < \kappa_f < \dfrac{c}{2\eta_{l_{\max}}}$, the assumption $f_i \,\forall i \in [N]$ are $\kappa_f$-accurate in expectation.

**Theorem 8.** *Let the objective function for the $i$-th device $f_i$ be $L$-smooth and convex, and Assumption 8 holds. For $c > \frac{1}{2} + \kappa_f \eta_{l_{\max}}$, SGD with ARMIJO Line search (3) achieves the convergence rate of deterministic gradient descent in expectation as*

$$\mathbb{E}\left[f_i(\bar{w}_k) - \inf f_i\right] \leq \frac{\tilde{c}}{(2\tilde{c}-1)\eta_{l_{\max}}K}\|w_0 - w^*\|^2$$

*where $\tilde{c} := c - 2\kappa_f \eta_{l_{\max}}$ and $\bar{w}_k = \frac{1}{K}\sum_{k=1}^{K} w_{k-1}$.*

*Proof.* Let $w^i_k$ be the iterate at the $k$-th step for a device $i$ running SGD update and $\eta_l$ is the learning rate returned by ARMIJO line search condition 3

$$\|w^i_k - w^*\|^2 = \|w^i_{k-1} - w^*\|^2 - 2\langle \eta^i_k \, g_i(w^i_{k-1}), w^i_{k-1} - w^* \rangle + \|\eta^i_k g_i(w^i_{k-1})\|^2$$

Taking the expectation on both sides conditioned on filtration $\mathcal{F}_{k-1}$

$$\mathbb{E}\left[\|w^i_k - w^*\|^2|\mathcal{F}_{k-1}\right] \leq \|w^i_{k-1} - w^*\|^2 + \underbrace{2\mathbb{E}\left[\langle \eta^i_k \, g_i(w^i_{k-1}), w^* - w^i_{k-1} \rangle|\mathcal{F}_{k-1}\right]}_{Term\ 1} + \mathbb{E}\left[(\eta^i_k)^2\|g_i(w^i_{k-1})\|^2|\mathcal{F}_{k-1}\right]$$

We first handle Term 1 as

$$\mathbb{E}\left[\langle \eta^i_k \, g_i(w^i_{k-1}), w^* - w^i_{k-1} \rangle|\mathcal{F}_{k-1}\right] = \mathbb{E}\left[\langle \eta_l \, g_i(w^i_{k-1}), w^* - w^i_{k-1} \rangle \mathbb{1}_{\{\langle g_i(w^i_{k-1}), w^* - w^i_{k-1}\rangle > 0\}}|\mathcal{F}_{k-1}\right]$$

$$+ \mathbb{E}\left[\langle \eta^i_k \, g_i(w^i_{k-1}), w^* - w^i_{k-1} \rangle \mathbb{1}_{\{\langle g_i(w^i_{k-1}), w^* - w^i_{k-1}\rangle \leq 0\}}|\mathcal{F}_{k-1}\right]$$

$$\leq \mathbb{E}\left[\langle \eta^i_k \, g_i(w^i_{k-1}), w^* - w^i_{k-1} \rangle \mathbb{1}_{\{\langle g_i(w^i_{k-1}), w^* - w^i_{k-1}\rangle > 0\}}|\mathcal{F}_{k-1}\right]$$

$$\leq \eta_{l_{\max}}\mathbb{E}\left[\langle g_i(w^i_{k-1}), w^* - w^i_{k-1} \rangle|\mathcal{F}_{k-1}\right]$$

$$= \eta_{l_{\max}}\langle \nabla f_i(w^i_{k-1}), w^* - w^i_{k-1} \rangle \tag{10}$$

Using Equation 10 and convexity of $f_i$, we obtain

$$\mathbb{E}\left[\|w^i_k - w^*\|^2|\mathcal{F}_{k-1}\right] \leq \|w^i_k - w^*\|^2 - 2\eta_{l_{\max}}(f_i(w^i_{k-1}) - \inf f_i) + \mathbb{E}\left[(\eta^i_k)^2\|g_i(w^i_k)\|^2|\mathcal{F}_{k-1}\right] \tag{11}$$

Since $\eta^i_k$ is returned by ARMIJO line search condition 3, thus it satisfies

$$f_i(w^i_k, \xi_k) - f_i(w^i_{k-1}, \xi_k) \leq -c\eta^i_k\|g_i(w^i_k)\|^2 \tag{12}$$

Rearranging and taking expectation on both sides of equation 12 conditioned on $\mathcal{F}_{k-1}$

$$\mathbb{E}\left[(\eta^i_k)^2\|g_i(w^i_k)\|^2|\mathcal{F}_{k-1}\right] \leq \frac{1}{c}\mathbb{E}\left[\eta^i_k(f_i(w^i_{k-1}, \xi_k) - f_i(w^i_k, \xi_k))|\mathcal{F}_{k-1}\right]$$

$$\leq \frac{\eta_{l_{\max}}}{c}\mathbb{E}\left[(f_i(w^i_{k-1}, \xi_k) - f_i(w^i_k, \xi_k))|\mathcal{F}_{k-1}\right]$$

$$\leq \frac{\eta_{l_{\max}}}{c}\mathbb{E}\left[(f_i(w^i_{k-1}, \xi_k) - f_i(w^i_{k-1}) + f_i(w^i_{k-1}) - f_i(w^i_k, \xi_k) + f_i(w^i_k) - f_i(w^i_k))|\mathcal{F}_{k-1}\right]$$

$$\leq \frac{\eta_{l_{\max}}}{c}\mathbb{E}\left[f_i(w^i_{k-1}) - f_i(w^i_k)) + 2\kappa_f\,(\eta^i_k)^2\|g_i(w^i_{k-1})\|^2|\mathcal{F}_{k-1}\right]$$

$$\tag{13}$$

where the last inequality is obtained using assumption 8. Rearranging the terms, we obtain

$$\left(1 - \frac{2\kappa_f \eta_{l_{\max}}}{c}\right) \mathbb{E}\left[(\eta_k^i)^2 \|g_i(w_k^i)\|^2 | \mathcal{F}_{k-1}\right] \leq \frac{\eta_{l_{\max}}}{c} \mathbb{E}\left[f_i(w_{k-1}^i) - f_i(w_k^i))|\mathcal{F}_{k-1}\right]$$

Choosing $\kappa_f < \frac{c}{2\eta_{l_{\max}}}$,

$$\mathbb{E}\left[(\eta_k^i)^2 \|g_i(w_k^i)\|^2 | \mathcal{F}_{k-1}\right] \leq \frac{\eta_{l_{\max}}}{c - 2\kappa_f \eta_{l_{\max}}} \mathbb{E}\left[f_i(w_{k-1}^i) - f_i(w_k^i))|\mathcal{F}_{k-1}\right] \qquad (14)$$

Substituting equation 14 in equation 11, we obtain

$$\mathbb{E}\left[\|w_k^i - w^*\|^2 | \mathcal{F}_{k-1}\right] \leq \|w_{k-1}^i - w^*\|^2 - 2\eta_{l_{\max}}(f_i(w_{k-1}^i) - \inf f_i) + \frac{\eta_{l_{\max}}}{c - 2\kappa_f \eta_{l_{\max}}} \mathbb{E}\left[(f_i(w_{k-1}^i) - f_i(w_k^i))|\mathcal{F}_{k-1}\right]$$

$$\leq \|w_k^i - w^*\|^2 - (f_i(w_k^i) - \inf f_i)\left(2\eta_{l_{\max}} - \frac{\eta_{l_{\max}}}{c - 2\kappa_f \eta_{l_{\max}}}\right)$$

Rearranging and summation on both sides for $k = 1, \dots, K$ and taking expectation again on both sides

$$\mathbb{E}\left[f_i(\bar{w}_k) - \inf f_i\right] \leq \frac{\tilde{c}}{(2\tilde{c} - 1)\eta_{l_{\max}} K} \|w_0 - w^*\|^2, \qquad (15)$$

for $c > \frac{1}{2} + \kappa_f \eta_{l_{\max}}$, $\tilde{c} := c - 2\kappa_f \eta_{l_{\max}}$ and $\bar{w}_k = \frac{1}{K} \sum_{k=1}^{K} w_{k-1}$. $\qquad \square$

Comparing Equations 9 and 15, we can see that the ARMIJO condition subsumes the effect of gradient noise. Moreover, as seen previously, under the interpolation condition, SGD behaves like GD. Thus, ARMIJO allows SGD to behave like GD without the interpolation condition. Similarly, we can recover a linear rate for strongly convex objectives using the definition of $\mu$-strongly convex objectives ($\mu > 0$) for SGD updates implemented with ARMIJO line-search.

Now, we discuss the case for non-convex objectives.

**Theorem 9.** *Let the objective function for the $i$-th device $f_i$ be $L$-smooth and non-convex, and Assumption 8 holds. For $c > 2\kappa_f \eta_{l_{\max}}$, SGD with ARMIJO Line search 3 achieves the convergence rate of deterministic gradient descent in expectation as*

$$\min_{k \in [K]} \mathbb{E}\|\nabla f_i(w_{k-1}^i)\|^2 \leq \left(\frac{1}{\eta_{l_{\max}}} + \frac{L}{2\tilde{c}}\right) \frac{1}{K}((f_i(w_0) - f_i(w^*))$$

*where $\tilde{c} := c - 2\kappa_f \eta_{l_{\max}}$.*

*Proof.* Using the definition of $L$-smoothness

$$f_i(w_k^i) - f_i(w_{k-1}^i) \leq -\eta_k^i \langle \nabla f_i(w_{k-1}^i), g_i(w_{k-1}^i)\rangle + \frac{L}{2}\|\eta_k^i g_i(w_{k-1}^i)\|^2$$

Taking expectation on both sides, conditioned on $\mathcal{F}_{k-1}$

$$\mathbb{E}\left[f_i(w_k^i) - f_i(w_{k-1}^i)|\mathcal{F}_{k-1}\right] \leq \underbrace{-\mathbb{E}\left[\eta_k^i \langle \nabla f_i(w_{k-1}^i), g_i(w_{k-1}^i)\rangle|\mathcal{F}_{k-1}\right]}_{Term\ 1} + \frac{L}{2}\mathbb{E}\left[(\eta_k^i)^2\|g_i(w_{k-1}^i)\|^2|\mathcal{F}_{k-1}\right]$$

We first handle Term 1 as

$$-\mathbb{E}\left[\eta_k^i \langle \nabla f_i(w_{k-1}^i), g_i(w_{k-1}^i)\rangle|\mathcal{F}_{k-1}\right] = -\mathbb{E}\left[\eta_k^i \langle \nabla f_i(w_{k-1}^i), g_i(w_{k-1}^i)\rangle \mathbb{1}_{\{\langle \nabla f_i(w_{k-1}^i), g_i(w_{k-1}^i)\rangle > 0\}}|\mathcal{F}_{k-1}\right]$$

$$- \mathbb{E}\left[\eta_k^i \langle \nabla f_i(w_{k-1}^i), g_i(w_{k-1}^i)\rangle \mathbb{1}_{\{\langle \nabla f_i(w_{k-1}^i), g_i(w_{k-1}^i)\rangle \leq 0\}}|\mathcal{F}_{k-1}\right]$$

$$\leq -\mathbb{E}\left[\eta_k^i \langle \nabla f_i(w_{k-1}^i), g_i(w_{k-1}^i)\rangle \mathbb{1}_{\{\langle \nabla f_i(w_{k-1}^i), g_i(w_{k-1}^i)\rangle \leq 0\}}|\mathcal{F}_{k-1}\right]$$

$$= -\mathbb{E}\left[\eta_k^i \langle \nabla f_i(w_{k-1}^i), g_i(w_{k-1}^i)\rangle|\mathcal{F}_{k-1}\right] \qquad (16)$$

$$\leq -\eta_{l_{\max}}\|\nabla f_i(w_{k-1}^i)\|^2 \qquad (17)$$

Notice that Equation 16 holds true when $\nabla f_i(w_{k-1}^i), g_i(w_{k-1}^i)\rangle \leq 0$. The last inequality is true since $\langle \nabla f_i(w_{k-1}^i), g_i(w_{k-1}^i)\rangle \leq 0$ in Equation 16. Using Equation 17, we obtain

$$\mathbb{E}\left[f_i(w_k^i) - f_i(w_{k-1}^i)|\mathcal{F}_{k-1}\right] \leq -\eta_{l_{\max}}\|f_i(w_{k-1}^i)\|^2 + \frac{L}{2}\mathbb{E}\left[(\eta_k^i)^2\|g_i(w_{k-1}^i)\|^2|\mathcal{F}_{k-1}\right] \quad (18)$$

Since $\eta_k^i$ is returned by ARMIJO line search condition 3, thus it satisfies

$$f_i(w_k^i, \xi_k) - f_i(w_{k-1}^i, \xi_k) \leq -c\eta_k^i\|g_i(w_{k-1}^i)\|^2 \quad (19)$$

Rearranging and taking expectation on both sides of equation 19

$$\mathbb{E}\left[(\eta_k^i)^2\|g_i(w_{k-1}^i)\|^2|\mathcal{F}_{k-1}\right] \leq \frac{1}{c}\mathbb{E}\left[\eta_k^i(f_i(w_{k-1}^i, \xi_k) - f_i(w_k^i, \xi_k))|\mathcal{F}_{k-1}\right]$$

$$\leq \frac{\eta_{l_{\max}}}{c}\mathbb{E}\left[(f_i(w_{k-1}^i) - f_i(w_k^i)) + 2\kappa_f(\eta_k^i)^2\|g_i(w_{k-1}^i)\|^2|\mathcal{F}_{k-1}\right]$$

For $\kappa_f < \frac{c}{2\eta_{l_{\max}}}$

$$\mathbb{E}\left[(\eta_k^i)^2\|g_i(w_{k-1}^i)\|^2|\mathcal{F}_{k-1}\right] \leq \frac{\eta_{l_{\max}}}{c - 2\kappa_f\eta_{l_{\max}}}\mathbb{E}\left[(f_i(w_{k-1}^i) - f_i(w_k^i))|\mathcal{F}_{k-1}\right] \quad (20)$$

Substituting equation 20 in equation 18, we obtain

$$\mathbb{E}\left[f_i(w_k^i) - f_i(w_{k-1}^i)|\mathcal{F}_{k-1}\right] \leq -\eta_{l_{\max}}\|f_i(w_{k-1}^i)\|^2 + \frac{L\eta_{l_{\max}}}{2(c - 2\kappa_f\eta_{l_{\max}})}\mathbb{E}\left[(f_i(w_{k-1}^i) - f_i(w_k^i))|\mathcal{F}_{k-1}\right]$$

Rearranging and putting $\tilde{c} := c - 2\kappa_f\eta_{l_{\max}}$

$$\|\nabla f_i(w_{k-1}^i)\|^2 \leq \left(\frac{1}{\eta_{l_{\max}}} + \frac{L}{2\tilde{c}}\right)\mathbb{E}\left[(f_i(w_k^i) - f_i(w_{k+1}^i)|w_k^i\right]$$

Summation on $k = 1, \ldots, K$ on both sides and taking expectations again

$$\sum_{k\in[K]}\mathbb{E}\|\nabla f_i(w_{k-1}^i)\|^2 \leq \left(\frac{1}{\eta_{l_{\max}}} + \frac{L}{2\tilde{c}}\right)\mathbb{E}\left[(f_i(w_0) - f_i(w_K^i)\right]$$

$$\min_{k\in[K]}\mathbb{E}\|\nabla f_i(w_{k-1}^i)\|^2 \leq \left(\frac{1}{\eta_{l_{\max}}} + \frac{L}{2\tilde{c}}\right)\frac{1}{K}((f_i(w_0) - f_i(w^*)) \quad (21)$$

for $\tilde{c} > 0$. $\qquad\qquad\square$

Thus, we can see that for objective classes of convex and non-convex objectives, using the ARMIJO line-search technique mitigates the effect of variance and the convergence rate for SGD is improved to match its deterministic counterpart in expectation. Motivated by this insight, we use ARMIJO line search in Federated Learning to mitigate the effect of client drift and gradient noise.

# B    MODEL UPDATE ALGORITHMS FOR FEDERATED LEARNING

## B.1    FEDSLS WITH ARMIJO LINE SEARCH

We describe the algorithm for FEDSLS- as run by a server orchestrating $N$ clients in Algorithm 4. Server initializes the global model $w_0$ and sends it to all clients. A random subset of $S$ clients is selected in each global communication round. The local model of each participating client is initialized to the current global model, and each client runs a local optimizer for $K$ rounds. Step 8 of FEDSLS (Algorithm 4) calls SGD-ARMIJO method (see Algorithm 2), which essentially uses ARMIJO line-search for SGD updates at each client for each $k$-th round. After $K$ local rounds, the pseudogradient $\Delta_{t,i}$ for each client is computed and sent to the server to obtain a global pseudogradient $\Delta_t$, which is then used for a gradient step-like update at the server to evaluate the next global model $w_{t+1}$.

## B.2    FEDEXPSLS WITH ARMIJO LINE SEARCH

We now describe the FEDEXPSLS Algorithm 5. In FEDEXPSLS, each participating client calls SGD−ARMIJO line-search, and the global model is updated using **server-side extrapolated learning rate** (Jhunjhunwala et al., 2023) computed using squared norms of local and global pseudogradients.

---

**Algorithm 4** FEDSLS

---

**Server Input**: initial global estimate $w_0$, total $N$ clients, sampled clients $\mathcal{S}_t$, where $|\mathcal{S}_t| = S$, batch size $b$, maximum bound for local learning rate $\eta_{l_{max}}$, global step-size $\eta_{g_t} = \eta_g \leq 1$

**Output**: global model update $w_T$

  1: **for** synchronization round $t = 0, 1, \ldots, T-1$ **do**
  2:     server sends $w_t$ to all clients
  3:     $\mathcal{S}_t \leftarrow$ random set of $S$ clients
  4:     **for** each $i \in \mathcal{S}_t$ in parallel **do**
  5:         $w_{t,0}^i \leftarrow w_t$
  6:         **for** $k = 1, 2, \ldots, K$ **do**
  7:            $w_{t,k}^i \leftarrow \text{SGD}-\text{ARMIJO}(w_{t,k-1}^i, \eta, \eta_{l_{max}}, \delta, b, k, opt)$
  8:         **end for**
  9:         $\Delta_t^i \leftarrow w_t - w_{t,K}^i$
10:     **end for**
11:     $\Delta_t \leftarrow \frac{1}{S} \sum_{i \in \mathcal{S}_t} \Delta_{t,i}$
12:     $w_{t+1} \leftarrow (w_t - \eta_{g_t} \Delta_t)$
13: **end for**
14: **return** $w_T$

---

---

**Algorithm 5** FEDEXPSLS algorithm.

---

**Server Input**: initial global estimate $w_0$, total $N$ clients, sampled clients $\mathcal{S}_t$, where $|\mathcal{S}_t| = S$, batch size $b$, maximum bound for local learning rate $\eta_{l_{max}}$, global step-size $\eta_{g_t} = \eta_g \leq 1$

**Output**: global model update $w_T$

  1: **for** each round $t = 0, 1, 2, \ldots, T-1$ **do**
  2:     Server sends $w_t$ to all clients
  3:     $\mathcal{S}_t \leftarrow$ random set of $S$ clients;
  4:     **for** each client $i \in \mathcal{S}_t$ in parallel **do**
  5:         $w_{t,0}^i \leftarrow w_t$
  6:         **for** local round $k = 1, 2, \ldots, K$ **do**
  7:            $w_{t,k}^i \leftarrow \text{SGD}-\text{ARMIJO}(w_{t,k-1}^i, \eta, \eta_{l_{max}}, \delta, b, k, opt)$
  8:         **end for**
  9:         $\Delta_t^i \leftarrow w_t - w_{t,K}^i$
10:     **end for**
11:     $\Delta_t = \frac{1}{S} \sum_{i \in \mathcal{S}_t} \Delta_t^i$
12:     $\eta_{g_t} \leftarrow \max \left\{ 1, \frac{\sum_{i \in \mathcal{S}_t} \|\Delta_t^i\|^2}{2|\mathcal{S}_t|(\|\Delta_t\|^2 + \epsilon)} \right\}$
13:     $w_{t+1} \leftarrow w_t - \eta_{g_t} \Delta_t$
14: **end for**
15: **return** $w_T$

---

## C   Proofs for FEDSLS

We can't apply the law of iterated expectations for reducing the Armijo line for stochastic functions to formulate an Armijo condition for the true function at that iterate. This is because the use of the same minibatch to evaluate the iterate and the function value at that iterate to check if the Armijo condition is satisfied. This gives the motivation that function estimate at a sample is actually a biased estimate of the true function. Thus, we adopted the notion of expected sufficiently accurate function estimates.

**Lemma 3.** *Under assumption 6, there exists $c' := (c - 2\kappa_f \eta_{\max}) > 0$, equivalently, $\kappa_f < \frac{c}{2\eta_{l_{\max}}}$, such that* ARMIJO *line-search (3) yields an expected decrease in the local objective $f_i$,*

$$\mathbb{E}\left[ \frac{1}{S} \sum_{i \in \mathcal{S}_t} \left( f_i(w_{t,K}^i) - f_i(w_t) \right) \right] \leq -\frac{c'}{S} \mathbb{E}\left[ \sum_{k, i \in \mathcal{S}_t} (\eta_{t,k}^i)^2 \|g_i(w_{t,k-1}^i)\|^2 \mid \mathcal{F}_t \right] \qquad (22)$$

*Proof.* Consider, the ARMIJO line-search (3) for some $c > 0$ as given below:

$$f_i(w_{t,k}^i, \xi_k) - f_i(w_{t,k-1}^i, \xi_k) \le -c\eta_{t,k}^i \|g_i(w_{t,k-1}^i)\|^2$$
$$f_i(w_{t,k}^i) - f_i(w_{t,k-1}^i) \le -c\eta_{t,k}^i \|g_i(w_{t,k-1}^i)\|^2 + (f_i(w_{t,k}^i) - f_i(w_{t,k}^i, \xi_k)) + (f_i(w_{t,k-1}^i, \xi_k) - f_i(w_{t,k-1}^i))$$

Summation over $k \in [K]$ and averaging over $i \in \mathcal{S}_t$ and taking expectations on both sides conditioned on filtration $\mathcal{F}_t$ gives

$$\mathbb{E}\left[\frac{1}{S}\sum_{k,i\in\mathcal{S}_t}\left(f_i(w_{t,k}^i) - f_i(w_{t,k-1}^i)\right)\Big|\mathcal{F}_t\right] \le -c\mathbb{E}\left[\frac{1}{S}\sum_{k,i\in\mathcal{S}_t}\eta_{t,k}^i\|g_i(w_{t,k-1}^i)\|^2 \mid \mathcal{F}_t\right]$$

$$+ \mathbb{E}\left[\frac{1}{S}\sum_{k,i\in\mathcal{S}_t}\left(f_i(w_{t,k}^i) - f_i(w_{t,k}^i, \xi_k)\right) \mid \mathcal{F}_t\right]$$

$$+ \mathbb{E}\left[\frac{1}{S}\sum_{k,i\in\mathcal{S}_t}\left(f_i(w_{t,k-1}^i, \xi_k) - f_i(w_{t,k-1}^i)\right) \mid \mathcal{F}_t\right]$$

$$\overset{\text{Asm 6}}{\le} -c\mathbb{E}\left[\frac{1}{S}\sum_{k,i\in\mathcal{S}_t}\eta_{t,k}^i\|g_i(w_{t,k-1}^i)\|^2 \mid \mathcal{F}_t\right]$$

$$+ 2\kappa_f\mathbb{E}\left[\frac{1}{S}\sum_{k,i\in\mathcal{S}_t}(\eta_{t,k}^i)^2\|g_i(w_{t,k-1}^i)\|^2 \mid \mathcal{F}_t\right]$$

$$\le -\frac{(c - 2\kappa_f\eta_{l_{\max}})}{S}\mathbb{E}\left[\sum_{k,i\in\mathcal{S}_t}(\eta_{t,k}^i)^2\|g_i(w_{t,k-1}^i)\|^2 \mid \mathcal{F}_t\right]$$

where $c' := (c - 2\kappa_f\eta_{l_{\max}}) > 0$, when $\kappa_f < \frac{c}{2\eta_{l_{\max}}}$. $\square$

**Lemma 4.** *Under the assumption 7 and 6, the model drift from the global update $w_t$ to local updates per client after $K$ local steps $w_{t,K}^i$ across all clients $i$ for the* FEDSLS *algorithm is bounded as*

$$\sum_i \mathbb{E}\left[\|w_t - w_{t,K}^i\|^2\right] \le \frac{\eta_{l_{\max}}NK}{c'}\mathbb{E}\left[(f(w_t) - f(w^*))\right], \tag{23}$$

*where $c' > 0$.*

*Proof.* Using Lemma 3, we obtain

$$\mathbb{E}\left[\frac{1}{S}\sum_{i\in\mathcal{S}_t}\left(f_i(w_{t,K}^i) - f_i(w_t)\right)\Big|\mathcal{F}_t\right] \le -\frac{c'}{S}\mathbb{E}\left[\sum_{k,i\in\mathcal{S}_t}\eta_{t,k}^i\|g_i(w_{t,k-1}^i)\|^2\Big|\mathcal{F}_t\right]$$

$$\le -\frac{c'}{N}\sum_{k,i}\mathbb{E}\left[\eta_{t,k}^i\|g_i(w_{t,k-1}^i)\|^2\big|\mathcal{F}_t\right]$$

$$\le -\frac{c'}{N}\sum_{k,i}\mathbb{E}\left[\frac{1}{\eta_{t,k}^i}\|\eta_{t,k}^i g_i(w_{t,k-1}^i)\|^2\Big|\mathcal{F}_t\right]$$

$$\le -\frac{c'}{\eta_{l_{\max}}N}\sum_{k,i}\mathbb{E}\left[\|w_{t,k-1}^i - w_{t,k}^i\|^2\big|\mathcal{F}_t\right]$$

$$\le -\frac{c'}{\eta_{l_{\max}}N}\sum_{k,i}\mathbb{E}\left[\|w_{t,k-1}^i - w_{t,k}^i\|^2\big|\mathcal{F}_t\right],$$

where we used $\eta_{t,k}^i \leq \eta_{l_{\max}}$. Expanding over $k = 1, \ldots, K$ on right-hand side

$$\mathbb{E}\left[\frac{1}{S}\sum_{i\in\mathcal{S}_t}\left(f_i(w_{t,K}^i) - f_i(w_t)\right)\Big|\mathcal{F}_t\right] \leq -\frac{c'}{\eta_{l_{\max}}N}\mathbb{E}\left[\sum_i\left(\|w_t - w_{t,1}^i\|^2 + \|w_{t,1}^i - w_{t,2}^i\|^2\right.\right.$$
$$\left.\left. + \ldots + \|w_{t,K-1}^i - w_{t,K}^i\|^2\right)\Big|\mathcal{F}_t\right]$$
$$\leq -\frac{c'}{\eta_{l_{\max}}NK}\sum_i\mathbb{E}\left[\|w_t - w_{t,K}^i\|^2\big|\mathcal{F}_t\right]$$

Last inequality is obtained using the fact $\|w_t - w_{t,K}^i\|^2 \leq K\sum_k\|w_{t,k-1}^i - w_{t,k}^i\|^2$. We obtain

$$\sum_i\mathbb{E}\left[\|w_t - w_{t,K}^i\|^2\big|\mathcal{F}_t\right] \leq \frac{\eta_{l_{\max}}NK}{c'}\mathbb{E}\left[\frac{1}{S}\sum_{i\in\mathcal{S}_t}\left(f_i(w_t) - f_i(w_{t,K}^i)\right)\Big|\mathcal{F}_t\right]$$

Under assumption 7 on $f_i(w^*) \leq f_i(w)$ for all $w \in \mathbb{R}^d$, we obtain

$$\sum_i\mathbb{E}\left[\|w_t - w_{t,K}^i\|^2\big|\mathcal{F}_t\right] \leq \frac{\eta_{l_{\max}}NK}{c'}\mathbb{E}\left[\frac{1}{S}\sum_{i\in\mathcal{S}_t}\left(f_i(w_t) - f_i(w^*)\right)\Big|\mathcal{F}_t\right]$$
$$\leq \frac{\eta_{l_{\max}}NK}{c'}\mathbb{E}\left[\left(f(w_t) - f(w^*)\right)\Big|\mathcal{F}_t\right] \tag{24}$$

Taking the expectation again gives the result. $\qquad\square$

**Lemma 5.** *Under assumptions 7 and 6, the updates of* FEDSLS *have bounded drift using* ARMIJO *line-search*

$$\frac{1}{N}\sum_{i,k}\mathbb{E}\left[\|w_t - w_{t,k-1}^i\|^2\right] \leq \frac{\eta_{l_{\max}}K^2}{c'}\mathbb{E}\left[\left(f(w_t) - f(w^*)\right)\right].$$

*Proof.* Recall that the local update made on client $i$ is $w_{t,k}^i = w_{t,k-1}^i - \eta_{t,k}^i g_i(w_{t,k-1}^i)$, where $\eta_{t,k}^i$ is obtained using ARMIJO line-search. Thus,

$$\frac{1}{N}\sum_{i,k}\|w_t - w_{t,k-1}^i\|^2 = \frac{1}{N}\sum_{i,k}\left\|\sum_{j=1}^{k-1}\eta_{t,j}^i g_i(w_{t,j-1}^i)\right\|^2$$
$$\leq \frac{1}{N}\sum_{i,k}(k-1)\sum_{j=1}^{k-1}\left(\eta_{t,j}^i\right)^2\left\|g_i(w_{t,j-1}^i)\right\|^2. \tag{25}$$

Using ARMIJO rule, we have $\frac{c}{\eta_{t,k}^i}\left\|\eta_{t,k}^i g_i(w_{t,k-1}^i)\right\|^2 \leq f_i(w_{t,k-1}^i, \xi_k^i) - f_i(w_{t,k}^i, \xi_k^i)$, thus we can write

$$c\left(\eta_{t,k}^i\right)^2\left\|g_i(w_{t,k-1}^i)\right\|^2 \leq \eta_{t,k}^i\left(f_i(w_{t,k-1}^i, \xi_k^i) - f_i(w_{t,k}^i, \xi_k^i)\right)$$
$$\leq \eta_{l_{\max}}\left(\left(f_i(w_{t,k-1}^i, \xi_k^i) - f_i(w_{t,k-1}^i)\right) - \left(f_i(w_{t,k}^i, \xi_k^i) - f_i(w_{t,k}^i)\right)\right.$$
$$\left. + \left(f_i(w_{t,k-1}^i) - f_i(w_{t,k}^i)\right)\right). \tag{26}$$

Taking expectation on both sides of Equation (26) conditioned on $\mathcal{F}_t$ and using assumption 6, we obtain

$$c\mathbb{E}\left[\left(\eta_{t,k}^i\right)^2\left\|g_i(w_{t,k-1}^i)\right\|^2\Big|\mathcal{F}_t\right] \leq \eta_{l_{\max}}\left(\mathbb{E}\left[f_i(w_{t,k-1}^i, \xi_k^i) - f_i(w_{t,k-1}^i)\Big|\mathcal{F}_t\right] - \mathbb{E}\left[f_i(w_{t,k}^i, \xi_k^i) - f_i(w_{t,k}^i)\Big|\mathcal{F}_t\right]\right.$$
$$\left. + \mathbb{E}\left[f_i(w_{t,k-1}^i) - f_i(w_{t,k}^i)\Big|\mathcal{F}_t\right]\right)$$
$$\leq 2\kappa_f\eta_{l_{\max}}\mathbb{E}\left[\left(\eta_{t,k}^i\right)^2\left\|g_i(w_{t,k-1}^i)\right\|^2\Big|\mathcal{F}_t\right] + \eta_{l_{\max}}\mathbb{E}\left[f_i(w_{t,k-1}^i) - f_i(w_{t,k}^i)\Big|\mathcal{F}_t\right].$$

Thus, for $c' > 0$ i.e., $\kappa_f < \frac{c}{2\eta_{l_{\max}}}$, we have

$$\mathbb{E}\left[\left(\eta_{t,k}^i\right)^2\left\|g_i(w_{t,k-1}^i)\right\|^2\Big|\mathcal{F}_t\right] \leq \frac{\eta_{l_{\max}}}{c'}\mathbb{E}\left[f_i(w_{t,k-1}^i) - f_i(w_{t,k}^i)\Big|\mathcal{F}_t\right], \tag{27}$$

Taking expectation on both sides of Equation (25) conditioned on $\mathcal{F}_t$ and substituting Equation (27)

$$\frac{1}{N}\sum_{i,k}\mathbb{E}\left[\|w_t - w_{t,k-1}^i\|^2\Big|\mathcal{F}_t\right] \le \frac{1}{N}\sum_{i,k}(k-1)\sum_{j=1}^{k-1}\mathbb{E}\left[\left(\eta_{t,j}^i\right)^2\left\|g_i(w_{t,j-1}^i)\right\|^2\Big|\mathcal{F}_t\right]$$

$$\le \frac{\eta_{l_{\max}}}{c'N}\sum_{i,k}(k-1)\sum_{j=1}^{k-1}\mathbb{E}\left[f_i(w_{t,j-1}^i) - f_i(w_{t,j}^i)\Big|\mathcal{F}_t\right]$$

$$\le \frac{\eta_{l_{\max}}}{c'N}\sum_{i,k}(k-1)\mathbb{E}\left[f_i(w_t) - f_i(w_{t,k-1}^i)\Big|\mathcal{F}_t\right].$$

Using the descent property of ARMIJO line-search $\mathbb{E}[f_i(w_{t,k-1}^i)|\mathcal{F}_t] \ge \mathbb{E}[f_i(w_{t,K}^i)|\mathcal{F}_t]$ for all $k$ from Lemma 3, we obtain

$$\frac{1}{N}\sum_{i,k}\mathbb{E}\left[\|w_t - w_{t,k-1}^i\|^2\Big|\mathcal{F}_t\right] \le \frac{\eta_{l_{\max}}K^2}{c'N}\sum_i\mathbb{E}\left[f_i(w_t) - f_i(w_{t,K}^i)\Big|\mathcal{F}_t\right]$$

Under assumption 7, the inequality becomes

$$\frac{1}{N}\sum_{i,k}\mathbb{E}\left[\|w_t - w_{t,k-1}^i\|^2\Big|\mathcal{F}_t\right] \le \frac{\eta_{l_{\max}}K^2}{c'}\mathbb{E}\left[f(w_t) - f(w^*)\Big|\mathcal{F}_t\right].$$

$\square$

## C.1 PROOF FOR CONVEX OBJECTIVES

We now give the convergence proof for convex functions.

**Theorem 10** (Restatement from Section 4: For constant server step-size). *Let the functions $f_i$ satisfy the assumptions 1, 2, 7 and 6. For a constant global learning rate $\eta_{g_t} = \eta_g$ and client learning rate $\eta_{l_{\max}} < \frac{2c-\eta_g}{(KL+4\kappa_f)}$, FEDSLS achieves the convergence rate for average of iterates as*

$$\mathbb{E}[f(\bar{w}_t) - f(w^*)] \le \frac{c'}{(2c' - \eta_g - KL\eta_{l_{\max}})\eta_g\eta_{l_{\max}}KT}\|w_0 - w^*\|^2 \tag{28}$$

*where $\bar{w}_t = \frac{1}{T}\sum_{t=0}^{T-1}w_t$ and $c' := c - 2\kappa_f\eta_{l_{\max}}$, such that $c' > 0$.*

*Proof.*

$$\|w_{t+1} - w^*\|^2 = \|w_t - \eta_g\Delta_t - w^*\|^2$$
$$= \|w_t - w^*\|^2 + \eta_g^2\|\Delta_t\|^2 - 2\eta_g\langle\Delta_t, w_t - w^*\rangle$$

Taking the expectation on both sides

$$\mathbb{E}[\|w_{t+1} - w^*\|^2\big|\mathcal{F}_t] = \|w_t - w^*\|^2 + \underbrace{\eta_g^2\mathbb{E}\left[\|\Delta_t\|^2\big|\mathcal{F}_t\right]}_{\mathcal{A}_1} + \underbrace{2\eta_g\mathbb{E}[\langle\Delta_t, w^* - w_t\rangle|\mathcal{F}_t]}_{\mathcal{A}_2} \tag{29}$$

We first resolve $\mathcal{A}_1$ by using lemma 4 under interpolation regime,

$$\mathcal{A}_1 = \eta_g^2\mathbb{E}\left[\|\Delta_t\|^2\big|\mathcal{F}_t\right]$$

$$= \eta_g^2\mathbb{E}\left[\left\|\frac{1}{S}\sum_{i\in\mathcal{S}_t}(w_t - w_{t,K}^i)\right\|^2\Big|\mathcal{F}_t\right]$$

$$\le \eta_g^2\mathbb{E}\left[\frac{1}{S}\sum_{i\in\mathcal{S}_t}\left\|(w_t - w_{t,K}^i)\right\|^2\big|\mathcal{F}_t\right]$$

$$\le \frac{\eta_g^2}{N}\sum_i\mathbb{E}\left[\left\|(w_t - w_{t,K}^i)\right\|^2\big|\mathcal{F}_t\right]$$

$$\le \frac{\eta_g^2\eta_{l_{\max}}K}{c'}\mathbb{E}\left[f(w_t) - f(w^*)\big|\mathcal{F}_t\right], \tag{30}$$

where $c' := (c - 2\kappa_f \eta_{l_{\max}}) > 0$. We now resolve $\mathcal{A}_2$,

$$\mathcal{A}_2 = 2\eta_g \mathbb{E}[\langle \Delta_t, w^* - w_t \rangle | \mathcal{F}_t]$$

$$= 2\eta_g \mathbb{E}\left[\left\langle \frac{1}{S} \sum_{k,i \in \mathcal{S}_t} \eta_{t,k}^i g_i(w_{t,k-1}^i), w^* - w_t \right\rangle \Big| \mathcal{F}_t\right]$$

$$= \frac{2\eta_g}{N} \sum_{k,i} \mathbb{E}\left[\langle \eta_{t,k}^i g_i(w_{t,k-1}^i), w^* - w_t \rangle | \mathcal{F}_t\right]$$

$$= \frac{2\eta_g}{N} \sum_{k,i} \mathbb{E}\left[\eta_{t,k}^i \langle g_i(w_{t,k-1}^i), w^* - w_t \rangle \mathbb{1}_{\{\langle g_i(w_{t,k-1}^i), w^* - w_t \rangle \geq 0\}} \Big| \mathcal{F}_t\right]$$

$$+ \frac{2\eta_g}{N} \sum_{k,i} \mathbb{E}\left[\eta_{t,k}^i \langle g_i(w_{t,k-1}^i), w^* - w_t \rangle \mathbb{1}_{\{\langle g_i(w_{t,k-1}^i), w^* - w_t \rangle < 0\}} \Big| \mathcal{F}_t\right]$$

$$= \frac{2\eta_g}{N} \sum_{k,i} \eta_{l_{\max}} \mathbb{E}\left[\langle g_i(w_{t,k-1}^i), w^* - w_t \rangle \mathbb{1}_{\{\langle g_i(w_{t,k-1}^i), w^* - w_t \rangle \geq 0\}} \Big| \mathcal{F}_t\right]$$

$$+ \frac{2\eta_g}{N} \sum_{k,i} \mathbb{E}\left[\eta_{t,k}^i \langle g_i(w_{t,k-1}^i), w^* - w_t \rangle \mathbb{1}_{\{\langle g_i(w_{t,k-1}^i), w^* - w_t \rangle < 0\}} \Big| \mathcal{F}_t\right]$$

Now since,

$$\eta_{t,k}^i \langle g_i(w_{t,k-1}^i), w^* - w_t \rangle \mathbb{1}_{\eta_{t,k}^i \langle g_i(w_{t,k-1}^i), w^* - w_t \rangle \leq 0} \leq 0$$

$$\implies \mathbb{E}[\eta_{t,k}^i \langle g_i(w_{t,k-1}^i), w^* - w_t \rangle \mathbb{1}_{\eta_{t,k}^i \langle g_i(w_{t,k-1}^i), w^* - w_t \rangle \leq 0} | \mathcal{F}_t] \leq \mathbb{E}[0 | \mathcal{F}_t]$$

So, we have

$$\mathcal{A}_2 = 2\eta_g \mathbb{E}[\langle \Delta_t, w^* - w_t \rangle | \mathcal{F}_t]$$

$$\leq 2\eta_g \frac{1}{N} \sum_{i \in [N]} \sum_{k=1}^K \eta_{l_{\max}} \mathbb{E}[\langle g_i(w_{t,k-1}^i), w^* - w_t \rangle \mathbb{1}_{\{\langle g_i(w_{t,k-1}^i), w^* - w_t \rangle \geq 0\}} | \mathcal{F}_t]$$

$$\leq 2\eta_g \frac{1}{N} \sum_{i \in [N]} \sum_{k=1}^K \eta_{l_{\max}} \mathbb{E}[\langle g_i(w_{t,k-1}^i), w^* - w_t \rangle | \mathcal{F}_t]$$

where the last inequality is due to the fact that- $\langle g_i(w_{t,k-1}^i), w^* - w_t \rangle \mathbb{1}_{\{\langle g_i(w_{t,k-1}^i), w^* - w_t \rangle \geq 0\}} \leq \langle g_i(w_{t,k-1}^i), w^* - w_t \rangle$ as indicator function $\mathbb{1}_{\{.\}} \leq 1$. Thus, we can now bound $\mathcal{A}_2$ as

$$\mathcal{A}_2 = 2\eta_g \mathbb{E}[\langle \Delta_t, w^* - w_t \rangle | \mathcal{F}_t]$$

$$\leq 2\eta_g \frac{1}{N} \sum_{i \in [N]} \sum_{k=1}^K \eta_{l_{\max}} \mathbb{E}[\langle g_i(w_{t,k-1}^i), w^* - w_t \rangle | \mathcal{F}_t]$$

$$= -\frac{2\eta_g \eta_{l_{\max}}}{N} \left\langle \sum_{i,k} \mathbb{E}[\nabla f_i(w_{t,k-1}^i) | \mathcal{F}_t], w_t - w^* \right\rangle$$

$$= -\frac{2\eta_g \eta_{l_{\max}}}{N} \sum_{i,k} \mathbb{E}[\langle \nabla f_i(w_{t,k-1}^i), w_t - w_{t,k-1}^i + w_{t,k-1}^i - w^* \rangle | \mathcal{F}_t]$$

$$= \frac{2\eta_g \eta_{l_{\max}}}{N} \sum_{i,k} \mathbb{E}[-\langle \nabla f_i(w_{t,k-1}^i), w_{t,k-1}^i - w^* \rangle + \langle \nabla f_i(w_{t,k-1}^i), w_{t,k-1}^i - w_t \rangle | \mathcal{F}_t]$$

$$\overset{\text{convexity}}{\leq} \frac{2\eta_g \eta_{l_{\max}}}{N} \sum_{i,k} \mathbb{E}[(f_i(w^*) - f_i(w_{t,k-1}^i)) + \langle \nabla f_i(w_{t,k-1}^i), w_{t,k-1}^i - w_t \rangle | \mathcal{F}_t]$$

$$\overset{\text{smoothness}}{\leq} \frac{2\eta_g \eta_{l_{\max}}}{N} \sum_{i,k} \mathbb{E}[(f_i(w^*) - f_i(w_{t,k-1}^i)) + (f_i(w_{t,k-1}^i) - f_i(w_t) + \frac{L}{2}\|w_t - w_{t,k-1}^i\|^2) | \mathcal{F}_t]$$

$$\leq 2\eta_g \eta_{l_{\max}} K \mathbb{E}[(f(w^*) - f(w_t)) | \mathcal{F}_t] + \frac{\eta_g \eta_{l_{\max}} L}{N} \sum_{i,k} \mathbb{E}[\|w_t - w_{t,k-1}^i\|^2 | \mathcal{F}_t] \tag{31}$$

Substituting Equations (30) and (31) in Equation (29) and taking expectations on both sides

$$\mathbb{E}[\|w_{t+1} - w^*\|^2] \le \mathbb{E}[\|w_t - w^*\|^2] + \frac{\eta_g^2 \eta_{l_{\max}} K}{c'} \mathbb{E}\left[f(w_t) - f(w^*)\right] + 2\eta_g \eta_{l_{\max}} K \mathbb{E}[(f(w^*) - f(w_t))]$$

$$+ \frac{\eta_g \eta_{l_{\max}} L}{N} \sum_{i,k} \mathbb{E}[\|w_t - w_{t,k-1}^i\|^2] \tag{32}$$

Using Lemma 5

$$\mathbb{E}[\|w_{t+1} - w^*\|^2] \le \mathbb{E}[\|w_t - w^*\|^2] + \frac{\eta_g^2 \eta_{l_{\max}} K}{c'} \mathbb{E}\left[f(w_t) - f(w^*)\right] - 2\eta_g \eta_{l_{\max}} K \mathbb{E}[(f(w_t) - f(w^*))]$$

$$+ \frac{\eta_g \eta_{l_{\max}}^2 K^2 L}{c'} \mathbb{E}\left[(f(w_t) - f(w^*))\right] \tag{33}$$

$$\mathbb{E}[\|w_{t+1} - w^*\|^2] \le \mathbb{E}[\|w_t - w^*\|^2] - \eta_g \eta_{l_{\max}} K \mathbb{E}\left[f(w_t) - f(w^*)\right] \left(2 - \frac{\eta_g}{c'} - \frac{1}{c'} K \eta_{l_{\max}} L\right) \tag{34}$$

Rearranging the terms and assuming $\eta_{l_{\max}} < \frac{2c' - \eta_g}{KL}$, we obtain

$$\left(\frac{2c' - \eta_g - KL\eta_{l_{\max}}}{c'}\right) \eta_g \eta_{l_{\max}} K \mathbb{E}[f(w_t) - f(w^*)] \le \mathbb{E}[\|w_t - w^*\|^2] - \mathbb{E}[\|w_{t+1} - w^*\|^2]$$

Averaging over $t = 0, \dots, T - 1$ and using Jensen's inequality

$$\mathbb{E}[f(\bar{w}_t) - f(w^*)] \le \frac{c'}{(2c' - \eta_g - KL\eta_{l_{\max}})\eta_g \eta_{l_{\max}} KT} \mathbb{E}[(\|w_0 - w^*\|^2 - \|w_T - w^*\|^2)]$$

$$\le \frac{c'}{(2c' - \eta_g - KL\eta_{l_{\max}})\eta_g \eta_{l_{\max}} KT} \|w_0 - w^*\|^2, \tag{35}$$

where $\bar{w}_t = \frac{1}{T} \sum_{t=0}^{T-1} w_t$. $\qquad\square$

## C.2 PROOF FOR STRONGLY CONVEX OBJECTIVES

The proof for strongly convex functions follows similarly to the proof for convex objectives.

**Theorem 11** (Restatement from Section 4). *Let the functions $f_i$ satisfy the assumptions 1, 3, 7 and 6. For a constant global learning rate $\eta_{g_t} = \eta_g$ such that $\eta_g \le \frac{2}{\eta_{l_{\max}} \mu K}$, client learning rate $\eta_{l_{\max}} < \frac{2c - \eta_g}{KL + 4\kappa_f}$, FEDSLS algorithm satisfies*

$$\mathbb{E}\|w_T - w^*\|^2 \le \left(1 - \frac{\eta_g \eta_{l_{\max}} \mu K}{2}\right)^T \|w_0 - w^*\|^2.$$

*Proof.*

$$\|w_{t+1} - w^*\|^2 = \|w_t - \eta_g \Delta_t - w^*\|^2$$
$$= \|w_t - w^*\|^2 + \eta_g^2 \|\Delta_t\|^2 - 2\eta_g \langle \Delta_t, w_t - w^* \rangle$$

Taking the expectation on both sides

$$\mathbb{E}[\|w_{t+1} - w^*\|^2 | \mathcal{F}_t] = \|w_t - w^*\|^2 + \underbrace{\eta_g^2 \mathbb{E}\left[\|\Delta_t\|^2 | \mathcal{F}_t\right]}_{\mathcal{B}_1} + \underbrace{2\eta_g \mathbb{E}[\langle \Delta_t, w^* - w_t \rangle | \mathcal{F}_t]}_{\mathcal{B}_2}$$

We first resolve $\mathcal{B}_1$ by using Lemma 4 for $c' := c - 2\kappa_f \eta_{l_{\max}} > 0$,

$$\mathcal{B}_1 = \eta_g^2 \mathbb{E}\left[\|\Delta_t\|^2 | \mathcal{F}_t\right] = \eta_g^2 \mathbb{E}\left[\left\|\frac{1}{S} \sum_{i \in \mathcal{S}_t} (w_t - w_{t,K}^i)\right\|^2 | \mathcal{F}_t\right]$$

$$\le \eta_g^2 \frac{1}{N} \sum_i \mathbb{E}\left[\|(w_t - w_{t,K}^i)\|^2 | \mathcal{F}_t\right]$$

$$\le \frac{\eta_g^2 \eta_{l_{\max}} K}{c'} \mathbb{E}\left[f(w_t) - f(w^*) | \mathcal{F}_t\right] \tag{36}$$

We now resolve $\mathcal{B}_2$ using perturbed strong convexity (Karimireddy et al., 2020) using $\mu \leq L$

$$\mathcal{B}_2 = 2\eta_g \mathbb{E}[\langle \Delta_t, w^* - w_t \rangle | \mathcal{F}_t] \leq -\frac{2\eta_g \eta_{l_{\max}}}{N} \left\langle \sum_{i,k} \mathbb{E}[\nabla f_i(w_{t,k-1}^i)|\mathcal{F}_t], w_t - w^* \right\rangle$$

$$= -\frac{2\eta_g \eta_{l_{\max}}}{N} \sum_{i,k} \mathbb{E}[\langle \nabla f_i(w_{t,k-1}^i), w_t - w_{t,k-1}^i + w_{t,k-1}^i - w^* \rangle | \mathcal{F}_t]$$

$$= \frac{2\eta_g \eta_{l_{\max}}}{N} \sum_{i,k} \mathbb{E}[-\langle \nabla f_i(w_{t,k-1}^i), w_{t,k-1}^i - w^* \rangle + \langle \nabla f_i(w_{t,k-1}^i), w_{t,k-1}^i - w_t \rangle | \mathcal{F}_t]$$

$$\overset{\text{Using Asm 3}}{\leq} \frac{2\eta_g \eta_{l_{\max}}}{N} \sum_{i,k} \mathbb{E}[(f_i(w^*) - f_i(w_{t,k-1}^i)) - \frac{\mu}{2}\|w_{t,k-1}^i - w^*\|^2$$

$$+ \langle \nabla f_i(w_{t,k-1}^i), w_{t,k-1}^i - w_t \rangle | \mathcal{F}_t]$$

$$\overset{\text{smoothness}}{\leq} \frac{2\eta_g \eta_{l_{\max}}}{N} \sum_{i,k} \mathbb{E}[(f_i(w^*) - f_i(w_{t,k-1}^i)) - \frac{\mu}{4}\|w_t - w^*\|^2 | \mathcal{F}_t]$$

$$+ \frac{2\eta_g \eta_{l_{\max}}}{N} \sum_{i,k} \mathbb{E}[(f_i(w_{t,k-1}^i) - f_i(w_t) + \frac{L+\mu}{2}\|w_t - w_{t,k-1}^i\|^2) | \mathcal{F}_t]$$

$$\leq 2\eta_g \eta_{l_{\max}} K \mathbb{E}[(f(w^*) - f(w_t)) | \mathcal{F}_t] - \frac{\eta_g \eta_{l_{\max}} \mu K}{2} \mathbb{E}[\|w_t - w^*\|^2 | \mathcal{F}_t]$$

$$+ \frac{2\eta_g \eta_{l_{\max}} L}{N} \sum_{i,k} \mathbb{E}[\|w_t - w_{t,k-1}^i\|^2 | \mathcal{F}_t] \tag{37}$$

Combining Equations (36) and (37) and taking expectations on both sides

$$\mathbb{E}[\|w_{t+1} - w^*\|^2] \leq \mathbb{E}[\|w_t - w^*\|^2] + \frac{\eta_g^2 \eta_{l_{\max}} K}{c'} \mathbb{E}[f(w_t) - f(w^*)] + 2\eta_g \eta_{l_{\max}} K \mathbb{E}[(f(w^*) - f(w_t))]$$

$$- \frac{\eta_g \eta_{l_{\max}} \mu K}{2} \mathbb{E}[\|w_t - w^*\|^2] + \frac{\eta_g \eta_{l_{\max}} L}{N} \sum_{i,k} \mathbb{E}[\|w_t - w_{t,k-1}^i\|^2] \tag{38}$$

Using Lemma 5

$$\mathbb{E}[\|w_{t+1} - w^*\|^2] \leq \left(1 - \frac{\eta_g \eta_{l_{\max}} \mu K}{2}\right) \mathbb{E}[\|w_t - w^*\|^2] + \frac{\eta_g^2 \eta_{l_{\max}} K}{c'} \mathbb{E}[f(w_t) - f(w^*)]$$

$$- 2\eta_g \eta_{l_{\max}} K \mathbb{E}[(f(w_t) - f(w^*))] + \frac{\eta_g K^2 \eta_{l_{\max}}^2 L}{c'} \mathbb{E}[(f(w_t) - f(w^*))]$$

$$\mathbb{E}[\|w_{t+1} - w^*\|^2] \leq \left(1 - \frac{\eta_g \eta_{l_{\max}} \mu K}{2}\right) \mathbb{E}[\|w_t - w^*\|^2] - \eta_g \eta_{l_{\max}} K \mathbb{E}[f(w_t) - f(w^*)]\left(2 - \frac{\eta_g}{c'} - \frac{K\eta_{l_{\max}} L}{c'}\right)$$

For $\eta_{l_{\max}} \leq \frac{2c' - \eta_g}{KL}$, the term $\frac{2c' - \eta_g - \eta_{l_{\max}} KL}{c'} \eta_g \eta_{l_{\max}} K \mathbb{E}[f(w_t) - f(w^*)]$ becomes non-negative, thus resulting bound is given as

$$\mathbb{E}[\|w_{t+1} - w^*\|^2] \leq \left(1 - \frac{\eta_g \eta_{l_{\max}} \mu K}{2}\right) \mathbb{E}[\|w_t - w^*\|^2]. \tag{39}$$

Recursion over $t = 0, \ldots, T-1$ under the assumption $\eta_g \leq \frac{2}{\eta_{l_{\max}} \mu K}$

$$\mathbb{E}[\|w_T - w^*\|^2] \leq \left(1 - \frac{\eta_g \eta_{l_{\max}} \mu K}{2}\right)^T \mathbb{E}[\|w_0 - w^*\|^2]$$

$\square$

## C.3  PROOF FOR NON-CONVEX OBJECTIVES

**Theorem 12** (Restatement from Section 4). *Let functions $f_i$ satisfy the assumptions 1, 7 and 6. For*

$$\eta_{l_{max}} \geq \frac{\frac{8c'}{2T-1}}{\eta_g^2 LK + \sqrt{(\eta_g^2 LK)^2 + \eta_g L^2 K^2 \frac{16c'}{2T-1}}}, \text{FEDSLS } achieves \ the \ convergence \ rate$$

$$\min_{t=0,\ldots,\,T-1} \mathbb{E}[\|\nabla f(w_t)\|^2] \leq \frac{2L(\eta_{\max} LK + \eta_g)}{c'} \mathbb{E}[f(w_0) - f(w^*)],$$

*where $c' := c - 2\kappa_f \eta_{l_{\max}} > 0$.*

*Proof.* Using the smoothness of $f$

$$f(w_{t+1}) \leq f(w_t) + \langle \nabla f(w_t), (w_{t+1} - w_t) \rangle + \frac{L}{2}\|w_{t+1} - w_t\|^2$$

Taking expectations on both sides conditioned on $\mathcal{F}_t$ and bounding the inner product term similar to the proof in convex cases, we obtain

$$\mathbb{E}[f(w_{t+1}) \mid \mathcal{F}_t] \leq f(w_t) - \eta_g \langle \nabla f(w_t), \mathbb{E}[\Delta_t \mid \mathcal{F}_t] \rangle + \frac{L\eta_g^2}{2}\mathbb{E}[\|\Delta_t\|^2 \mid \mathcal{F}_t]$$

$$\leq f(w_t) - \eta_g \eta_{\max} \left\langle \nabla f(w_t), \frac{1}{N}\sum_{i,k} \nabla f_i(w_{t,k-1}^i) \right\rangle + \frac{L\eta_g^2}{2}\mathbb{E}[\|\Delta_t\|^2 \mid \mathcal{F}_t]$$

$$\leq f(w_t) - \eta_g \eta_{\max} K \left\langle \nabla f(w_t), \frac{1}{NK}\sum_{i,k} \nabla f_i(w_{t,k-1}^i) - \nabla f(w_t) + \nabla f(w_t) \right\rangle + \frac{L\eta_g^2}{2}\mathbb{E}[\|\Delta_t\|^2 \mid \mathcal{F}_t]$$

$$\leq f(w_t) - \eta_g \eta_{\max} K \left\langle \nabla f(w_t), \frac{1}{NK}\sum_{i,k} \nabla f_i(w_{t,k-1}^i) - \nabla f(w_t) \right\rangle$$

$$- \eta_g \eta_{\max} K \|\nabla f(w_t)\|^2 + \frac{L\eta_g^2}{2}\mathbb{E}[\|\Delta_t\|^2 \mid \mathcal{F}_t]$$

$$\leq f(w_t) + \eta_g \eta_{\max} K \left\langle \nabla f(w_t), \frac{1}{NK}\sum_{i,k} \left(\nabla f_i(w_t) - \nabla f_i(w_{t,k-1}^i)\right) \right\rangle$$

$$- \eta_g \eta_{\max} K \|\nabla f(w_t)\|^2 + \frac{L\eta_g^2}{2}\mathbb{E}[\|\Delta_t\|^2 \mid \mathcal{F}_t]$$

$$\overset{\text{CS Inq.}}{\leq} f(w_t) + \eta_g \eta_{\max} K \|\nabla f(w_t)\| \left\| \frac{1}{NK}\sum_{i,k} \left(\nabla f_i(w_t) - \nabla f_i(w_{t,k-1}^i)\right) \right\|$$

$$- \eta_g \eta_{\max} K \|\nabla f(w_t)\|^2 + \frac{L\eta_g^2}{2}\mathbb{E}[\|\Delta_t\|^2 \mid \mathcal{F}_t]$$

$$\overset{\text{Young's Inq.}}{\leq} f(w_t) + \frac{\eta_g \eta_{\max} K}{2} \left\| \frac{1}{NK}\sum_{i,k} \left(\nabla f_i(w_t) - \nabla f_i(w_{t,k-1}^i)\right) \right\|^2$$

$$- \frac{\eta_g \eta_{\max} K}{2} \|\nabla f(w_t)\|^2 + \frac{L\eta_g^2}{2}\mathbb{E}[\|\Delta_t\|^2 \mid \mathcal{F}_t]$$

$$\overset{\text{Jensen's}}{\leq} f(w_t) + \frac{\eta_g \eta_{\max}}{2N} \sum_{i,k} \left\|\nabla f_i(w_t) - \nabla f_i(w_{t,k-1}^i)\right\|^2 - \frac{\eta_g \eta_{\max} K}{2} \|\nabla f(w_t)\|^2 + \frac{L\eta_g^2}{2}\mathbb{E}[\|\Delta_t\|^2 \mid \mathcal{F}_t]$$

$$\overset{\text{Using Asm 1}}{\leq} f(w_t) + \frac{\eta_g \eta_{\max} L^2}{2N} \sum_{i,k} \left\|w_t - w_{t,k-1}^i\right\|^2 - \frac{\eta_g \eta_{\max} K}{2} \|\nabla f(w_t)\|^2 + \frac{L\eta_g^2}{2}\mathbb{E}[\|\Delta_t\|^2 \mid \mathcal{F}_t]$$

Taking expectation on both sides and using Lemma 5,

$$\mathbb{E}[f(w_{t+1})] \leq \mathbb{E}[f(w_t)] + \frac{\eta_g \eta_{l_{\max}}{}^2 L^2 K^2}{2c'} \mathbb{E}\left[f(w_t) - f(w^*)\right] - \frac{\eta_g \eta_{l_{\max}} K}{2} \mathbb{E}[\|\nabla f(w_t)\|^2]$$
$$+ \frac{L\eta_g{}^2}{2N} \sum_i \mathbb{E}[\|w_t - w_{t,K}^i\|^2].$$

Now, we use Lemma 4 to obtain

$$\mathbb{E}[f(w_{t+1})] \leq \mathbb{E}[f(w_t)] + \frac{\eta_g \eta_{l_{\max}}{}^2 L^2 K^2}{2c'} \mathbb{E}\left[f(w_t) - f(w^*)\right] - \frac{\eta_g \eta_{l_{\max}} K}{2} \mathbb{E}[\|\nabla f(w_t)\|^2]$$
$$+ \frac{L\eta_g{}^2 \eta_{l_{\max}} K}{2c'} \mathbb{E}[f(w_t) - f(w^*)].$$

Subtracting $f(w^*)$ from both sides and rearranging the terms, we obtain

$$\frac{\eta_g \eta_{l_{\max}} K}{2} \mathbb{E}[\|\nabla f(w_t)\|^2] \leq (1 + D) \mathbb{E}[f(w_t) - f(w^*)] - \mathbb{E}\left[f(w_{t+1}) - f(w^*)\right], \quad (40)$$

where $D := \frac{\eta_g \eta_{l_{\max}} LK(\eta_{l_{\max}} LK + \eta_g)}{2c'}$.

To create a telescoping scoping sum on the RHS, we use artificial weights $\alpha_t$, following (Stich, 2019), such that $\alpha_t \left(1 + \frac{\eta_g \eta_{l_{\max}}{}^2 L^2 K^2 (\eta_{l_{\max}} LK + \eta_g)}{2c'}\right) = \alpha_{t-1}$, where $\alpha_{-1} = 1$. Thus, multiplying $\alpha_t$ on both sides of Equation 51, we obtain

$$\alpha_t \frac{\eta_g \eta_{l_{\max}} K}{2} \mathbb{E}[\|\nabla f(w_t)\|^2] \leq \alpha_{t-1} \mathbb{E}[f(w_t) - f(w^*)] - \alpha_t \mathbb{E}\left[f(w_{t+1}) - f(w^*)\right].$$

Summing on both sides from $t = 0, \ldots, T - 1$, we obtain

$$\sum_{t=0}^{T-1} \alpha_t \frac{\eta_g \eta_{l_{\max}} K}{2} \mathbb{E}[\|\nabla f(w_t)\|^2] \leq \alpha_{-1} \mathbb{E}[f(w_0) - f(w^*)] - \alpha_{T-1} \mathbb{E}\left[f(w_{t+1}) - f(w^*)\right].$$

Since, $-\alpha_{T-1} \mathbb{E}\left[f(w_T) - f(w^*)\right]$ is a negative term, it can be ignored. Now, using $\alpha_{-1} = 1$ and diving both sides by $\sum_{t=0}^{T-1} \alpha_t$, we obtain

$$\min_{t=0,1,\ldots,\ T-1} \mathbb{E}[\|\nabla f(w_t)\|^2] \leq \frac{1}{\sum_{t=0}^{T-1} \alpha_t} \sum_{t=0}^{T-1} \alpha_t \mathbb{E}[\|\nabla f(w_t)\|^2] \leq \frac{2}{\eta_g \eta_{l_{\max}} K \sum_{t=0}^{T-1} \alpha_t} \mathbb{E}[f(w_0) - f(w^*)].$$

To find a final upper bound for the LHS, we need to find a lower bound for $\sum_{t=0}^{T-1} \alpha_t$. We evaluate $\sum_{t=0}^{T-1} \alpha_t$ as

$$\sum_{t=0}^{T-1} \alpha_t = \frac{1}{(1+D)} \frac{1 - \left(\frac{1}{1+D}\right)^T}{1 - \left(\frac{1}{1+D}\right)} = \frac{1}{(D)} \left(1 - \left(\frac{1}{1+D}\right)^T\right). \quad (41)$$

Choosing $\eta_{l_{\max}}$ such that $\left(\frac{1}{1+D}\right)^T \leq \frac{1}{2} \iff T \geq \frac{\log(2)}{\log(1+D)}$ provides a suitable lower bound for $\sum_{t=0}^{T-1} \alpha_t$. Using the identity $\frac{1}{\log(1+x)} \leq \frac{1}{x} + \frac{1}{2}$ for $x > 0$, we note that

$$\frac{\log(2)}{\log(1+D)} \leq \frac{1}{\log(1+D)} \leq \frac{1}{D} + \frac{1}{2}$$

Thus, it is sufficient to choose $\eta_{l_{\max}}$ such that

$$\frac{1}{D} + \frac{1}{2} \leq T \iff D := \frac{\eta_g \eta_{l_{\max}} LK(\eta_{l_{\max}} LK + \eta_g)}{2c'} \geq \frac{2}{2T - 1}.$$

Ignoring the negative root of the quadratic inequality $(\eta_g L^2 K^2)\eta_{l_{\max}} + (\eta_g^2 LK)\eta_{l_{\max}} \geq \frac{4c'}{2T-1}$, the bound for $\eta_{l_{\max}}$ is obtained as

$$\eta_{l_{max}} \geq \frac{\dfrac{8c'}{2T-1}}{\eta_g^2 LK + \sqrt{(\eta_g^2 LK)^2 + \eta_g L^2 K^2 \dfrac{16c'}{2T-1}}}, \tag{42}$$

using the quadratic formula $x = \frac{-2c}{b+\sqrt{b^2-4ac}}$ for a quadratic equation $ax^2 + bx + c = 0$. Hence, for $\eta_{l_{max}}$ satisfying Equation 53, we have $\left(\frac{1}{1+D}\right)^T \leq \frac{1}{2}$, thus $\sum_{t=0}^{T-1} \alpha_t == \frac{1}{(D)}\left(1 - \left(\frac{1}{1+D}\right)^T\right) \geq \frac{1}{2D}$. Thus, choosing $\eta_{l_{max}}$ according to Equation 53 for $2T-1 > 0$ when $T \geq 1$, we have

$$\min_{t=0,1,\ldots,\,T-1} \mathbb{E}[\|\nabla f(w_t)\|^2] \leq \frac{4D}{\eta_g \eta_{l_{\max}} K} \mathbb{E}[f(w_0) - f(w^*)]. \tag{43}$$

$\square$

# D  PROOFS FOR FEDEXPSLS

## D.1  CONVERGENCE PROOF OF FEDEXPSLS ALGORITHM: CONVEX OBJECTIVES

**Theorem 13** (Restatement from Section 4). *Suppose a function $f_i$ satisfy assumption 1 2, 7 and 6. For global learning rate $\eta_{g_t}$ as computed in FEDEXPSLS constrained to lie in $[1, \eta_{g_{\max}}]$, client learning rate $\eta_{l_{\max}} < \frac{2c'-1}{KL+4\kappa_f}$, FEDEXPSLS achieves the convergence rate for average of iterates as*

$$\mathbb{E}\left[f(\bar{w}_t) - f(w^*)\right] \leq \frac{c'}{(2c' - \eta_{l_{\max}} KL - 1)\eta_{l_{\max}} \eta_{g_{\max}} KT}\|w_0 - w^*\|^2,$$

*where $\bar{w}_t = \frac{1}{T}\sum_{t=1}^{T} w_t$ and $c'$.*

*Proof.*

$$\|w_{t+1} - w^*\|^2 = \|w_t - \eta_{g_t}\Delta_t - w^*\|^2$$
$$= \|w_t - w^*\|^2 + \eta_{g_t}^2\|\Delta_t\|^2 - 2\eta_{g_t}\langle\Delta_t, w_t - w^*\rangle$$
$$= \|w_t - w^*\|^2 + \eta_{g_t}^2\|\Delta_t\|^2 + 2\eta_{g_t}\langle\Delta_t, w^* - w_t\rangle$$

Taking expectations on both sides conditioned on $\mathcal{F}_t$

$$\mathbb{E}[\|w_{t+1} - w^*\|^2 \mid \mathcal{F}_t] = \|w_t - w^*\|^2 + \underbrace{\mathbb{E}[\eta_{g_t}^2\|\Delta_t\|^2 \mid \mathcal{F}_t]}_{\mathcal{C}_1} + \underbrace{2\mathbb{E}[\eta_{g_t}\langle\Delta_t, w^* - w_t\rangle \mid \mathcal{F}_t]}_{\mathcal{C}_2} \tag{44}$$

First, we bound the term $\mathcal{C}_1$ as below

$$\mathcal{C}_1 = \mathbb{E}[\eta_{g_t}^2\|\Delta_t\|^2 \mid \mathcal{F}_t] = \mathbb{E}\left[\eta_{g_t}\max\left\{1, \frac{\sum_{i\in\mathcal{S}_t}\|\Delta_t^i\|^2}{2S(\|\Delta_t\|^2 + \varepsilon)}\right\}\|\Delta_t\|^2\Big|\mathcal{F}_t\right]$$

$$\leq \mathbb{E}\left[\eta_{g_t}\frac{\sum_{i\in\mathcal{S}_t}\|\Delta_t^i\|^2}{S\|\Delta_t\|^2}\|\Delta_t\|^2\Big|\mathcal{F}_t\right]$$

$$\leq \mathbb{E}\left[\frac{1}{S}\sum_{i\in\mathcal{S}_t}\eta_{g_t}\|\Delta_t^i\|^2\Big|\mathcal{F}_t\right]$$

$$\leq \frac{1}{N}\sum_{i\in[N]}\mathbb{E}\left[\eta_{g_t}\|\Delta_t^i\|^2\Big|\mathcal{F}_t\right] = \frac{1}{N}\sum_{i\in[N]}\mathbb{E}\left[\eta_{g_t}\|w_t - w_{t,K}^i\|^2\Big|\mathcal{F}_t\right]$$

$$\leq \frac{\eta_{l_{\max}}\eta_{g_{\max}}K}{c'}\mathbb{E}\left[(f(w_t) - f(w^*))\big|\mathcal{F}_t\right],$$

where the last inequality is obtained using Lemma 4 for $c' = c - 2\kappa_f \eta_{l_{\max}}$ and $\eta_{g_t} \le \eta_{g_{\max}}$. We now resolve $\mathcal{C}_2$ as in proof of Theorem 2 (using indicator functions)

$$\mathcal{C}_2 = -2\mathbb{E}[\langle \eta_{g_t} \Delta_t, w_t - w^* \rangle | \mathcal{F}_t]$$

$$\le -\frac{2\eta_{l_{\max}} \eta_{g_{\max}}}{N} \left\langle \sum_{i,k} \nabla f_i(w_{t,k-1}^i), w_t - w^* \right\rangle$$

$$= -\frac{2\eta_{l_{\max}} \eta_{g_{\max}}}{N} \sum_{i,k} \left\langle \nabla f_i(w_{t,k-1}^i), w_t - w_{t,k-1}^i + w_{t,k-1}^i - w^* \right\rangle$$

$$= \frac{2\eta_{l_{\max}} \eta_{g_{\max}}}{N} \sum_{i,k} \left\{ -\left\langle \nabla f_i(w_{t,k-1}^i), w_{t,k-1}^i - w^* \right\rangle + \left\langle \nabla f_i(w_{t,k-1}^i), w_{t,k-1}^i - w_t \right\rangle \right\}$$

$$\overset{\text{convexity}}{\le} \frac{2\eta_{l_{\max}} \eta_{g_{\max}}}{N} \sum_{i,k} \left\{ \left( f_i(w^*) - f_i(w_{t,k-1}^i) \right) + \left\langle \nabla f_i(w_{t,k-1}^i), w_{t,k-1}^i - w_t \right\rangle \right\}$$

$$\overset{\text{smoothness}}{\le} \frac{2\eta_{l_{\max}} \eta_{g_{\max}}}{N} \sum_{i,k} \left\{ \left( f_i(w^*) - f_i(w_{t,k-1}^i) \right) + \left( f_i(w_{t,k-1}^i) - f_i(w_t) + \frac{L}{2}\|w_t - w_{t,k-1}^i\|^2 \right) \right\}$$

$$\le \frac{2\eta_{l_{\max}} \eta_{g_{\max}}}{N} \sum_{i,k} \left( f_i(w^*) - f_i(w_t) \right) + \frac{\eta_{l_{\max}} \eta_{g_{\max}} L}{N} \sum_{i,k} \|w_t - w_{t,k-1}^i\|^2$$

$$\le \frac{2\eta_{l_{\max}} \eta_{g_{\max}}}{N} \sum_{i,k} \left( f_i(w^*) - f_i(w_t) \right) + \frac{\eta_{l_{\max}} \eta_{g_{\max}} L}{N} \sum_{i,k} \|w_t - w_{t,k-1}^i\|^2$$

$$\le 2\eta_{l_{\max}} \eta_{g_{\max}} K \left( f(w^*) - f(w_t) \right) + \frac{\eta_{l_{\max}} \eta_{g_{\max}} L}{N} \sum_{i,k} \|w_t - w_{t,k-1}^i\|^2 \tag{45}$$

Substituting the bounds on the terms $\mathcal{C}_1$ and $\mathcal{C}_2$ in Equation 44

$$\mathbb{E}[\|w_{t+1} - w^*\|^2 \mid \mathcal{F}_t] \le \|w_t - w^*\|^2 + \frac{\eta_{l_{\max}} \eta_{g_{\max}} K}{c'} \mathbb{E}\left[ f(w_t) - f(w^*) | \mathcal{F}_t \right]$$

$$+ 2\eta_{l_{\max}} \eta_{g_{\max}} K \left( f(w^*) - f(w_t) \right) + \frac{\eta_{l_{\max}} \eta_{g_{\max}} L}{N} \sum_{i,k} \|w_t - w_{t,k-1}^i\|^2 \tag{46}$$

Taking expectation again, using the tower property and substituting the bound on client drift across $N$ clients and $k$ local rounds using Lemma 5, we obtain

$$\mathbb{E}[\|w_{t+1} - w^*\|^2] \le \mathbb{E}[\|w_t - w^*\|^2] + \frac{\eta_{l_{\max}} \eta_{g_{\max}} K}{c'} \mathbb{E}\left[ f(w_t) - f(w^*) \right] + 2\eta_{l_{\max}} \eta_{g_{\max}} K \mathbb{E}\left[ f(w^*) - f(w_t) \right]$$

$$+ \frac{\eta_{l_{\max}}^2 \eta_{g_{\max}} K^2 L}{c'} \mathbb{E}\left[ f(w_t) - f(w^*) \right]$$

Rearranging, we obtain

$$\mathbb{E}[\|w_{t+1} - w^*\|^2] \le \mathbb{E}[\|w_t - w^*\|^2] - \eta_{l_{\max}} \eta_{g_{\max}} K \mathbb{E}\left[ f(w_t) - f(w^*) \right] \left( 2 - \frac{1}{c'} - \frac{\eta_{l_{\max}} K L}{c'} \right)$$

For $\eta_{l_{\max}} < \frac{2c'-1}{KL}$, we have

$$\frac{(2c' - \eta_{l_{\max}} KL - 1)}{c'} \eta_{l_{\max}} \eta_{g_{\max}} K \mathbb{E}\left[ f(w_t) - f(w^*) \right] \le \mathbb{E}[\|w_t - w^*\|^2] - \mathbb{E}[\|w_{t+1} - w^*\|^2] \tag{47}$$

Averaging over $t = 1, \ldots, T$ and using Jensen's inequality

$$\mathbb{E}\left[ f(\bar{w}_t) - f(w^*) \right] \le \frac{c'}{(2c' - \eta_{l_{\max}} KL - 1)\eta_{l_{\max}} \eta_{g_{\max}} KT} \|w_0 - w^*\|^2,$$

where $\bar{w}_t = \frac{1}{T} \sum_{t=1}^{T} w_t$. $\qquad\square$

## D.2 Convergence Proof of FedExpSLS Algorithm: Strongly-Convex Objectives

**Theorem 14** (Restatement from Section 4). *Let the functions $f_i$ satisfy assumption 1, 3, 7 and 6. For a global learning rate $\eta_{g_t}$ computed in* FEDEXPSLS *constrained to lie in $[1, \eta_{g_{\max}}]$ such that $\eta_{g_{\max}} \leq \frac{2}{\eta_{l_{\max}}\mu K}$, client learning rate $\eta_{l_{\max}} < \frac{2c-1}{KL+4\kappa_f}$, the last update of* FEDEXPSLS *satisfies*

$$\mathbb{E}\|w_{T+1} - w^*\|^2 \leq \left(1 - \frac{\eta_g \eta_{l_{\max}}\mu K}{2}\right)^{T+1}\|w_0 - w^*\|^2.$$

*Proof.*

$$\|w_{t+1} - w^*\|^2 = \|w_t - \eta_{g_t}\Delta_t - w^*\|^2$$
$$= \|w_t - w^*\|^2 + \eta_{g_t}^2\|\Delta_t\|^2 - 2\eta_{g_t}\langle\Delta_t, w_t - w^*\rangle$$
$$= \|w_t - w^*\|^2 + \eta_{g_t}^2\|\Delta_t\|^2 + 2\eta_{g_t}\langle\Delta_t, w^* - w_t\rangle$$

Taking expectations on both sides

$$\mathbb{E}[\|w_{t+1} - w^*\|^2 \mid \mathcal{F}_t] = \|w_t - w^*\|^2 + \underbrace{\mathbb{E}[\eta_{g_t}^2\|\Delta_t\|^2 \mid \mathcal{F}_t]}_{\mathcal{D}_1} + \underbrace{2\mathbb{E}[\eta_{g_t}\langle\Delta_t, w^* - w_t\rangle \mid \mathcal{F}_t]}_{\mathcal{D}_2} \quad (48)$$

First, we bound the term $\mathcal{D}_1$ same as in Theorem 6

$$\mathcal{D}_1 := \mathbb{E}[\eta_{g_t}^2\|\Delta_t\|^2 \mid \mathcal{F}_t] \leq \frac{\eta_{l_{\max}}\eta_{g_{\max}}K}{c'}\mathbb{E}\left[(f(w_t) - f(w^*)) \mid \mathcal{F}_t\right],$$

We now resolve $\mathcal{D}_2$ using perturbed strong convexity (Karimireddy et al., 2020) using $\mu \leq L$

$$\mathcal{D}_2 = -2\eta_{g_t}\mathbb{E}[\langle\Delta_t, w_t - w^*\rangle|\mathcal{F}_t]$$
$$\leq -\frac{2\eta_{l_{\max}}\eta_{g_{\max}}}{N}\left\langle\sum_{i,k}\mathbb{E}[\nabla f_i(w_{t,k-1}^i)|\mathcal{F}_t], w_t - w^*\right\rangle$$
$$= -\frac{2\eta_{l_{\max}}\eta_{g_{\max}}}{N}\sum_{i,k}\mathbb{E}[\langle\nabla f_i(w_{t,k-1}^i), w_t - w_{t,k-1}^i + w_{t,k-1}^i - w^*\rangle|\mathcal{F}_t]$$
$$= \frac{2\eta_{l_{\max}}\eta_{g_{\max}}}{N}\sum_{i,k}\mathbb{E}[-\langle\nabla f_i(w_{t,k-1}^i), w_{t,k-1}^i - w^*\rangle + \langle\nabla f_i(w_{t,k-1}^i), w_{t,k-1}^i - w_t\rangle|\mathcal{F}_t]$$
$$\overset{\text{Using Asm 3}}{\leq} \frac{2\eta_{l_{\max}}\eta_{g_{\max}}}{N}\sum_{i,k}\mathbb{E}[(f_i(w^*) - f_i(w_{t,k-1}^i)) - \frac{\mu}{2}\|w_{t,k-1}^i - w^*\|^2 + \langle\nabla f_i(w_{t,k-1}^i), w_{t,k-1}^i - w_t\rangle|\mathcal{F}_t]$$
$$\overset{\text{smoothness}}{\leq} \frac{2\eta_{l_{\max}}\eta_{g_{\max}}}{N}\sum_{i,k}\mathbb{E}[(f_i(w^*) - f_i(w_{t,k-1}^i)) - \frac{\mu}{4}\|w_t - w^*\|^2|\mathcal{F}_t]$$
$$+ \frac{2\eta_{l_{\max}}\eta_{g_{\max}}}{N}\sum_{i,k}\mathbb{E}[(f_i(w_{t,k-1}^i) - f_i(w_t) + \frac{L+\mu}{2}\|w_t - w_{t,k-1}^i\|^2)|\mathcal{F}_t]$$
$$\leq 2\eta_{l_{\max}}\eta_{g_{\max}}K\mathbb{E}[(f(w^*) - f(w_t))|\mathcal{F}_t] - \frac{\eta_{l_{\max}}\eta_{g_{\max}}\mu K}{2}\mathbb{E}[\|w_t - w^*\|^2|\mathcal{F}_t]$$
$$+ \frac{2\eta_{l_{\max}}\eta_{g_{\max}}L}{N}\sum_{i,k}\mathbb{E}[\|w_t - w_{t,k-1}^i\|^2|\mathcal{F}_t] \quad (49)$$

Substituting the bounds on the terms $\mathcal{D}_1$ and $\mathcal{D}_2$ in Equation 48

$$\mathbb{E}[\|w_{t+1} - w^*\|^2 \mid \mathcal{F}_t] \leq \|w_t - w^*\|^2 - \frac{\eta_{l_{\max}}\eta_{g_{\max}}\mu K}{2}\mathbb{E}[\|w_t - w^*\|^2|\mathcal{F}_t] + \frac{\eta_{l_{\max}}\eta_{g_{\max}}K}{c'}\mathbb{E}[f(w_t) - f(w^*)|\mathcal{F}_t]$$
$$+ 2\eta_{l_{\max}}\eta_{g_{\max}}K\mathbb{E}[(f(w^*) - f(w_t))|\mathcal{F}_t] + \frac{2\eta_{l_{\max}}\eta_{g_{\max}}L}{N}\sum_{i,k}\mathbb{E}[\|w_t - w_{t,k-1}^i\|^2|\mathcal{F}_t]$$

Taking expectation again, using tower property and substituting the bound on client drift across $N$ clients and $k$ local rounds using Lemma 5, we obtain

$$\mathbb{E}[\|w_{t+1} - w^*\|^2] \leq \left(1 - \frac{\eta_{l_{\max}}\eta_{g_{\max}}\mu K}{2}\right)\mathbb{E}[\|w_t - w^*\|^2] - \eta_{l_{\max}}\eta_{g_{\max}}K\mathbb{E}[(f(w_t) - f(w^*))]\left(2 - \frac{1}{c'} - \frac{\eta_{l_{\max}}KL}{c'}\right)$$

For $\eta_{l_{\max}} \leq \frac{2c'-1}{KL}$, the second term can be ignored. Hence, we obtain

$$\mathbb{E}[\|w_{t+1} - w^*\|^2] \leq \left(1 - \frac{\eta_{l_{\max}}\eta_{g_{\max}}\mu K}{2}\right)\mathbb{E}[\|w_t - w^*\|^2]$$

Recursion over $t = 0, \ldots, T-$ under the assumption $\eta_{g_{\max}} \leq \frac{2}{\eta_{l_{\max}}\mu K}$

$$\mathbb{E}[\|w_{T+1} - w^*\|^2] \leq \left(1 - \frac{\eta_{l_{\max}}\eta_{g_{\max}}\mu K}{2}\right)^{T+1}\|w_0 - w^*\|^2.$$

$\square$

## D.3 CONVERGENCE PROOF OF FEDEXPSLS ALGORITHM: NON- CONVEX OBJECTIVES

**Theorem 15.** *Let the functions $f_i$ satisfy assumption 1, 7 and 6. For a global learning rate $\eta_{g_t}$ computed in* FEDEXPSLS *constrained to lie in $[1, \eta_{g_{\max}}]$ and local learning rate bound $\eta_{l_{max}} \geq \frac{\frac{8c'}{2T-1}}{\eta_{g_{\max}}LK + \sqrt{(\eta_{g_{\max}}LK)^2 + \eta_{g_{\max}}L^2K^2\frac{16c'}{2T-1}}}$,* FEDSLS *achieves the convergence rate*

$$\min_{t=0,\ldots,\,T-1}\mathbb{E}[\|\nabla f(w_t)\|^2] \leq \frac{2L(\eta_{l_{\max}}LK + 1)}{c'}\mathbb{E}[f(w_0) - f(w^*)],$$

*where $c' := c - 2\kappa_f\eta_{l_{\max}} > 0$.*

*Proof.* Using the smoothness of $f$

$$f(w_{t+1}) \leq f(w_t) + \langle\nabla f(w_t), (w_{t+1} - w_t)\rangle + \frac{L}{2}\|w_{t+1} - w_t\|^2$$

Taking expectations on both sides conditioned on $\mathcal{F}_t$, we obtain

$$\mathbb{E}[f(w_{t+1}) \mid \mathcal{F}_t] \leq f(w_t) - \underbrace{\langle\nabla f(w_t), \mathbb{E}[\eta_{g_t}\Delta_t \mid \mathcal{F}_t]\rangle}_{\mathcal{T}_1} + \underbrace{\frac{L}{2}\mathbb{E}[\eta_{g_t}^2\|\Delta_t\|^2 \mid \mathcal{F}_t]}_{\mathcal{T}_2} \qquad (50)$$

We resolve $\mathcal{T}_2$ by bounding the inner product, similar to the proof in cases, and using $\eta_{g_t} \leq \eta_{g_{\max}}$ as

$$\mathcal{T}_1 := -\left\langle \nabla f(w_t), \mathbb{E}[\eta_{g_t}\Delta_t \mid \mathcal{F}_t]\right\rangle \leq -\eta_{g_{\max}}\eta_{l_{\max}}\left\langle \nabla f(w_t), \frac{1}{N}\sum_{i,k}\nabla f_i(w_{t,k-1}^i)\right\rangle$$

$$\leq -\eta_{g_{\max}}\eta_{l_{\max}}K\left\langle \nabla f(w_t), \frac{1}{NK}\sum_{i,k}\nabla f_i(w_{t,k-1}^i) - \nabla f(w_t) + \nabla f(w_t)\right\rangle$$

$$\leq -\eta_{g_{\max}}\eta_{l_{\max}}K\left\langle \nabla f(w_t), \frac{1}{NK}\sum_{i,k}\nabla f_i(w_{t,k-1}^i) - \nabla f(w_t)\right\rangle - \eta_{g_{\max}}\eta_{l_{\max}}K\|\nabla f(w_t)\|^2$$

$$\leq \eta_{g_{\max}}\eta_{l_{\max}}K\left\langle \nabla f(w_t), \frac{1}{NK}\sum_{i,k}\left(\nabla f_i(w_t) - \nabla f_i(w_{t,k-1}^i)\right)\right\rangle - \eta_{g_{\max}}\eta_{l_{\max}}K\|\nabla f(w_t)\|^2$$

$$\overset{\text{CS Inq.}}{\leq} \eta_{g_{\max}}\eta_{l_{\max}}K\left\|\nabla f(w_t)\right\|\left\|\frac{1}{NK}\sum_{i,k}\left(\nabla f_i(w_t) - \nabla f_i(w_{t,k-1}^i)\right)\right\| - \eta_{g_{\max}}\eta_{l_{\max}}K\|\nabla f(w_t)\|^2$$

$$\overset{\text{Young's Inq.}}{\leq} \frac{\eta_{g_{\max}}\eta_{l_{\max}}K}{2}\left\|\frac{1}{NK}\sum_{i,k}\left(\nabla f_i(w_t) - \nabla f_i(w_{t,k-1}^i)\right)\right\|^2 - \frac{\eta_{g_{\max}}\eta_{l_{\max}}K}{2}\|\nabla f(w_t)\|^2$$

$$\overset{\text{Jensen's}}{\leq} \frac{\eta_{g_{\max}}\eta_{l_{\max}}}{2N}\sum_{i,k}\left\|\nabla f_i(w_t) - \nabla f_i(w_{t,k-1}^i)\right\|^2 - \frac{\eta_{g_{\max}}\eta_{l_{\max}}K}{2}\|\nabla f(w_t)\|^2$$

$$\overset{\text{UsingAsm 1}}{\leq} \frac{\eta_{g_{\max}}\eta_{l_{\max}}L^2}{2N}\sum_{i,k}\left\|w_t - w_{t,k-1}^i\right\|^2 - \frac{\eta_{g_{\max}}\eta_{l_{\max}}K}{2}\|\nabla f(w_t)\|^2$$

Using Lemma 5,

$$\mathcal{T}_1 \leq \frac{\eta_{g_{\max}}\eta_{l_{\max}}^2 L^2 K^2}{2c'}\mathbb{E}\left[f(w_t) - f(w^*)\right] - \frac{\eta_{g_{\max}}\eta_{l_{\max}}K}{2}\mathbb{E}[\|\nabla f(w_t)\|^2]$$

We take expectation on both sides of Equation 50. Substituting the bound on $\mathcal{T}_1$ using Lemma 5 and bound on $\mathcal{T}_2$, we obtain

$$\mathbb{E}[f(w_{t+1})] \leq \mathbb{E}[f(w_t)] + \frac{\eta_{g_{\max}}\eta_{l_{\max}}^2 L^2 K^2}{2c'}\mathbb{E}\left[f(w_t) - f(w^*)\right] - \frac{\eta_{g_{\max}}\eta_{l_{\max}}K}{2}\mathbb{E}[\|\nabla f(w_t)\|^2]$$

$$+ \frac{L}{2N}\sum_i\mathbb{E}[\eta_{g_t}\|w_t - w_{t,K}^i\|^2].$$

Now, we use Lemma 4 to obtain

$$\mathbb{E}[f(w_{t+1})] \leq \mathbb{E}[f(w_t)] + \frac{\eta_{g_{\max}}\eta_{l_{\max}}^2 L^2 K^2}{2c'}\mathbb{E}\left[f(w_t) - f(w^*)\right] - \frac{\eta_{g_{\max}}\eta_{l_{\max}}K}{2}\mathbb{E}[\|\nabla f(w_t)\|^2]$$

$$+ \frac{L\eta_{g_{\max}}\eta_{l_{\max}}K}{2c'}\mathbb{E}[f(w_t) - f(w^*)].$$

Subtracting $f(w^*)$ from both sides and rearranging the terms, we obtain

$$\frac{\eta_{g_{\max}}\eta_{l_{\max}}K}{2}\mathbb{E}[\|\nabla f(w_t)\|^2] \leq \left(1 + \tilde{D}\right)\mathbb{E}[f(w_t) - f(w^*)] - \mathbb{E}\left[f(w_{t+1}) - f(w^*)\right], \qquad (51)$$

where $\tilde{D} := \dfrac{\eta_{g_{\max}}\eta_{l_{\max}}LK(\eta_{l_{\max}}LK + 1)}{2c'}$.

To create a telescoping scoping sum on the RHS, we use artificial weights $\beta_t$, following (Stich, 2019), such that $\beta_t\left(1 + \dfrac{\eta_{g_{\max}}\eta_{l_{\max}}^2 L^2 K^2(\eta_{l_{\max}}LK + 1)}{2c'}\right) = \beta_{t-1}$, where $\beta_{-1} = 1$. Thus, multiplying $\beta_t$ on both sides of Equation 51, we obtain

$$\beta_t\frac{\eta_{g_{\max}}\eta_{l_{\max}}K}{2}\mathbb{E}[\|\nabla f(w_t)\|^2] \leq \beta_{t-1}\mathbb{E}[f(w_t) - f(w^*)] - \beta_t\mathbb{E}\left[f(w_{t+1}) - f(w^*)\right].$$

Summing on both sides from $t = 0, \ldots, T - 1$, we obtain

$$\sum_{t=0}^{T-1} \beta_t \frac{\eta_{g_{\max}} \eta_{l_{\max}} K}{2} \mathbb{E}[\|\nabla f(w_t)\|^2] \leq \beta_{-1} \mathbb{E}[f(w_0) - f(w^*)] - \beta_{T-1} \mathbb{E}\left[f(w_{t+1}) - f(w^*)\right].$$

Since, $-\beta_{T-1} \mathbb{E}\left[f(w_T) - f(w^*)\right]$ is a negative term, it can be ignored. Now, using $\beta_{-1} = 1$ and diving both sides by $\sum_{t=0}^{T-1} \beta_t$, we obtain

$$\min_{t=0,1,\ldots,T-1} \mathbb{E}[\|\nabla f(w_t)\|^2] \leq \frac{1}{\sum_{t=0}^{T-1} \beta_t} \sum_{t=0}^{T-1} \beta_t \mathbb{E}[\|\nabla f(w_t)\|^2] \leq \frac{2}{\eta_g \eta_{l_{\max}} K \sum_{t=0}^{T-1} \beta_t} \mathbb{E}[f(w_0) - f(w^*)].$$

To find a final upper bound for the LHS, we need to find a lower bound for $\sum_{t=0}^{T-1} \beta_t$. We evaluate $\sum_{t=0}^{T-1} \beta_t$ as

$$\sum_{t=0}^{T-1} \beta_t = \frac{1}{(1 + \tilde{D})} \frac{1 - \left(\frac{1}{1+\tilde{D}}\right)^T}{1 - \left(\frac{1}{1+\tilde{D}}\right)} = \frac{1}{(\tilde{D})} \left(1 - \left(\frac{1}{1 + \tilde{D}}\right)^T\right). \tag{52}$$

Choosing $\eta_{l_{\max}}$ such that $\left(\frac{1}{1+\tilde{D}}\right)^T \leq \frac{1}{2} \iff T \geq \frac{\log(2)}{\log(1+\tilde{D})}$ provides a suitable lower bound for $\sum_{t=0}^{T-1} \beta_t$. Using the identity $\frac{1}{\log(1+x)} \leq \frac{1}{x} + \frac{1}{2}$ for $x > 0$, we note that

$$\frac{\log(2)}{\log(1 + \tilde{D})} \leq \frac{1}{\log(1 + \tilde{D})} \leq \frac{1}{\tilde{D}} + \frac{1}{2}$$

Thus, it is sufficient to choose $\eta_{l_{\max}}$ such that

$$\frac{1}{\tilde{D}} + \frac{1}{2} \leq T \iff \tilde{D} := \frac{\eta_{g_{\max}} \eta_{l_{\max}} LK(\eta_{l_{\max}} LK + 1)}{2c'} \geq \frac{2}{2T - 1}.$$

Ignoring the negative root of the quadratic inequality $(\eta_{g_{\max}} L^2 K^2)\eta_{l_{\max}} + (\eta_{g_{\max}} LK)\eta_{l_{\max}} \geq \frac{4c'}{2T-1}$, the bound for $\eta_{l_{\max}}$ is obtained as

$$\eta_{l_{max}} \geq \frac{\dfrac{8c'}{2T - 1}}{\eta_{g_{\max}} LK + \sqrt{(\eta_{g_{\max}} LK)^2 + \eta_{g_{\max}} L^2 K^2 \dfrac{16c'}{2T - 1}}}, \tag{53}$$

using the quadratic formula $x = \frac{-2c}{b + \sqrt{b^2 - 4ac}}$ for a quadratic equation $ax^2 + bx + c = 0$. Hence, for $\eta_{l_{max}}$ satisfying Equation 53, we have $\left(\frac{1}{1+\tilde{D}}\right)^T \leq \frac{1}{2}$, thus $\sum_{t=0}^{T-1} \beta_t == \frac{1}{(\tilde{D})} \left(1 - \left(\frac{1}{1+\tilde{D}}\right)^T\right) \geq \frac{1}{2\tilde{D}}$. Thus, choosing $\eta_{l_{max}}$ according to Equation 53 for $2T - 1 > 0$ when $T \geq 1$, we have

$$\min_{t=0,1,\ldots,T-1} \mathbb{E}[\|\nabla f(w_t)\|^2] \leq \frac{4\tilde{D}}{\eta_{g_{\max}} \eta_{l_{\max}} K} \mathbb{E}[f(w_0) - f(w^*)]. \tag{54}$$

$\square$

# E    COMPARISON OF ARMIJO LINE SEARCH WITH BOUNDED HETEROGENEITY

The maximum value of ARMIJO step-size for each client is fixed as $\eta_{l_{\max}}$, see Algorithm 2. Line-search for a client-local LR begins with $\eta_{l_{\max}}$ and continues until a maximally feasible $\eta_{t,k}^i$ is obtained that satisfies the Armijo condition.

We include Lemma 1 of Vaswani et al. (2019) for our discussion. Note that we do not use these bounds on the learning rate in our poofs.

**Lemma 6** (Lemma 1 of Vaswani et al. (2019)). *Assume that for each client $i$ and sample $\xi \sim \mathcal{D}_i$, the function $f_i(\cdot, \xi)$ is $L_\xi$-smooth (define, $L := \max_{\xi \sim \mathcal{D}} L_\xi$). Let $c \in (0,1)$ and $\eta_{l_{\max}} > 0$. At inner step $(t,k)$ on client $i$, the Armijo line search returns a step-size $\eta_{t,k}^i \in (0, \eta_{l_{\max}}]$ satisfying*

$$\eta_{t,k}^i \ \geq \ \min\left\{\frac{2(1-c)}{L}, \eta_{l_{\max}}\right\}.$$

*Proof.* Set $g_{t,k-1}^i := \nabla f_i(w_{t,k-1}^i, \xi_k)$ and $w_{t,k}^i := w_{t,k-1}^i - \eta_{t,k}^i g_{t,k-1}^i$. By $L$-smoothness,

$$f_i(w_{t,k}^i, \xi_k) \leq f_i(w_{t,k-1}^i, \xi_k) + \langle g_{t,k-1}^i, w_{t,k}^i - w_{t,k-1}^i\rangle + \frac{L}{2}\|w_{t,k}^i - w_{t,k-1}^i\|^2$$

$$= f_i(w_{t,k-1}^i, \xi_k) - \eta_{t,k}^i\|g_{t,k-1}^i\|^2 + \frac{L}{2}(\eta_{t,k}^i)^2\|g_{t,k-1}^i\|^2$$

$$= f_i(w_{t,k-1}^i, \xi_k) - \left(\eta_{t,k}^i - \frac{L(\eta_{t,k}^i)^2}{2}\right)\|g_{t,k-1}^i\|^2. \tag{55}$$

The Armijo condition with parameter $c > 0$ is

$$f_i(w_{t,k}^i, \xi_k) \ \leq \ f_i(w_{t,k-1}^i, \xi_k) - c\eta_{t,k}^i\|g_{t,k-1}^i\|^2. \tag{56}$$

A sufficient condition for equation 56 to hold is that $\left(\eta_{t,k}^i - \frac{L(\eta_{t,k}^i)^2}{2}\right)$ on the RHS of equation 55 dominates the Armijo decrease $c\eta_{t,k}^i$:

$$\eta_{t,k}^i - \frac{L(\eta_{t,k}^i)^2}{2} \ \geq \ c\eta_{t,k}^i \quad \Longleftrightarrow \quad \eta_{t,k}^i \ \leq \ \tau := \frac{2(1-c)}{L}.$$

Therefore, every $\eta \in \left(0, \min\{\tau, \eta_{l_{\max}}\}\right]$ is a feasible step that satisfies equation 56. Let us define the Armijo acceptance set at the inner step $(t,k)$ for a client $i$ as

$$\mathcal{A}_{t,k}^i := \left\{\eta \in (0, \eta_{l_{\max}}] : \text{ equation 56 holds}\right\}.$$

Clearly,

$$\left(0, \min\{\tau, \eta_{l_{\max}}\}\right] \ \subseteq \ \mathcal{A}_{t,k}^i.$$

By the line-search selection rule: return the *maximal* feasible step in $(0, \eta_{l_{\max}}]$,

$$\eta_{t,k}^i \ \geq \ \min\{\tau, \eta_{l_{\max}}\} \ = \ \min\left\{\frac{2(1-c)}{L}, \eta_{l_{\max}}\right\}$$

is returned by the line-search algorithm. $\qquad\square$

**Remarks.**

1. The inequality $\eta_{t,k}^i \leq \frac{2(1-c)}{L}$ is a sufficient condition for Armijo line search. Thus, $\left(0, \min\left\{\frac{2(1-c)}{L}, \eta_{l_{\max}}\right\}\right]$ is the guaranteed feasible set that will satisfy Armijo. The learning rate returned by Armijo, which is the maximal step-size such that equation 3 is satisfied, will be lower bounded by $\frac{2(1-c)}{L}$, hence $\frac{2(1-c)}{L}$ need not be the maximal step; larger steps are possible.

2. The lower bound concerns the *returned* step when the line-search selects the *largest feasible* step on its search set.

3. **Geometric backtracking** (Alg. 2, `opt=1`). If the search tests only the grid $\{\eta_{l_{\max}}, \beta\eta_{l_{\max}}, \beta^2\eta_{l_{\max}}, \ldots\}$ with fixed $\beta \in (0,1)$ and returns the largest grid point in $\mathcal{A}_{t,k}^i$, then $\eta_{t,k}^i \ \geq \ \beta \min\left\{\frac{2(1-c)}{L}, \eta_{l_{\max}}\right\}$.

   This is because if $\frac{2(1-c)}{L} \ \geq \ \eta_{l_{\max}}$, the search starts with $\eta_{l_{\max}}$ and the first test passes, so the returned step is $\eta = \eta_{l_{\max}} \geq \beta\eta_{l_{\max}} = \beta \min\left\{\frac{2(1-c)}{L}, \eta_{l_{\max}}\right\}$. Otherwise, if $\frac{2(1-c)}{L} < \eta_{l_{\max}}$, let $m$ be the smallest integer such that $\beta^m\eta_{l_{\max}} \leq \frac{2(1-c)}{L} < \beta^{m-1}\eta_{l_{\max}}$.

Since every $\eta \leq \frac{2(1-c)}{L}$ satisfies the Armijo condition, $\beta^m \eta_{l_{\max}}$ lies in the feasible set; hence the returned step satisfies $\eta \geq \beta^m \eta_{l_{\max}}$. Because $\frac{2(1-c)}{L} < \beta^{m-1}\eta_{l_{\max}}$, we have $\beta \frac{2(1-c)}{L} < \beta^m \eta_{l_{\max}} >$. Therefore $\eta \geq \beta^m \eta_{l_{\max}} > \beta \frac{2(1-c)}{L} = \beta \min\left\{\frac{2(1-c)}{L}, \eta_{l_{\max}}\right\}$.

For $\texttt{opt} \in \{0, 2\}$, the guarantee becomes $\eta_{t,k}^i \geq \beta \min\left\{\frac{2(1-c)}{L}, \eta_{\text{start}}\right\}$, where $\eta_{\text{start}}$ is the starting step-size used in reset. This can be arbitrarily small if $\eta_{\text{start}}$ is small.

For the analysis, we considered the search for step-size in the continuous space over all reals with $\texttt{opt=1}$, not the grid, i.e., the line search returns the largest feasible step in $(0, \eta_{l_{\max}}]$.

We now give the Lemma that provides an upper bound which allows ARMIJO to substitute the bounded heterogeneity assumption.

**Lemma 7.** *Under assumption 6, there exists $c' := (c - 2\kappa_f \eta_{\max}) > 0$, equivalently, $\kappa_f < \frac{c}{2\eta_{l_{\max}}}$, such that* ARMIJO *line-search (3) yields*

$$\sum_{k,i} \mathbb{E}\left[\|\nabla f_i(w_{t,k-1}^i)\|^2\right] \leq \max\left\{\frac{L}{2(1-c)}, \frac{1}{\eta_{l_{\max}}}\right\} \frac{1}{c'}\left(f(w_t) - \mathbb{E}\left[\sum_{i \in \mathcal{S}_t} \frac{1}{S} f_i(w_{t,K}^i)\right]\right).$$

*Proof.* Using Lemma 3, we obtain

$$\mathbb{E}\left[\frac{1}{S}\sum_{k,i\in\mathcal{S}_t}\left(f_i(w_{t,k}^i) - f_i(w_{t,k-1}^i)\right)\Big|\mathcal{F}_t\right] \overset{\text{Lemma 6}}{\leq} -\min\left\{\frac{2(1-c)}{L}, \eta_{l_{\max}}\right\}\frac{c'}{S}\mathbb{E}\left[\sum_{k,i\in\mathcal{S}_t}\|g_i(w_{t,k-1}^i)\|^2\Big|\mathcal{F}_t\right]$$

where $c' := (c - 2\kappa_f \eta_{l_{\max}}) > 0$, when $\kappa_f < \frac{c}{2\eta_{l_{\max}}}$. Using the squared mean as a lower bound for the second moment, we obtain

$$\mathbb{E}\left[\frac{1}{S}\sum_{k,i\in\mathcal{S}_t}\left(f_i(w_{t,k}^i) - f_i(w_{t,k-1}^i)\right)\Big|\mathcal{F}_t\right] \leq -\min\left\{\frac{2(1-c)}{L}, \eta_{l_{\max}}\right\}\frac{c'}{S}\mathbb{E}\left[\sum_{k,i\in\mathcal{S}_t}\|\nabla f_i(w_{t,k-1}^i)\|^2\Big|\mathcal{F}_t\right]$$

Thus, rearranging and expanding the telescoping sum, we obtain

$$\min\left\{\frac{2(1-c)}{L}, \eta_{l_{\max}}\right\}\frac{c'}{N}\sum_{k,i}\mathbb{E}\left[\|\nabla f_i(w_{t,k-1}^i)\|^2\big|\mathcal{F}_t\right] \leq \left(f(w_t) - \sum_{i\in\mathcal{S}_t}\frac{1}{S}\mathbb{E}\left[f_i(w_{t,K}^i)|\mathcal{F}_t\right]\right)$$

Thus, we have

$$\sum_{k,i}\mathbb{E}\left[\|\nabla f_i(w_{t,k-1}^i)\|^2\big|\mathcal{F}_t\right] \leq \max\left\{\frac{L}{2(1-c)}, \frac{1}{\eta_{l_{\max}}}\right\}\frac{1}{c'}\left(f(w_t) - \sum_{i\in\mathcal{S}_t}\frac{1}{S}\mathbb{E}\left[f_i(w_{t,K}^i)|\mathcal{F}_t\right]\right)$$

$\square$

# F EXTRA EXPERIMENTAL DETAILS

## Description of Dataset

**CIFAR10/100** The CIFAR-10 dataset is composed of 60,000 natural images of size 32×32 pixels, categorized into 10 distinct classes. CIFAR-100 builds on the same image set but introduces a more fine-grained classification scheme, dividing the images into 100 classes and thereby increasing the difficulty of the classification task. Both datasets consist of 50,000 training images and 10,000 test images. For training in the federated learning environment, the training data is artificially partitioned among 100 clients using the data partitioning strategy proposed by (Hsu et al., 2019), introducing non-IID characteristics across clients.

**FEMNIST** The FEMNIST data set is a federated variant of the EMNIST dataset, designed to benchmark personalized and federated learning algorithms as introduced by (Caldas et al., 2018), the dataset is naturally partitioned between 3,550 clients. The dataset contains a total of 80,5263 samples with 226.83 samples per user.

**SHAKESPEARE** The SHAKESPEARE dataset is a character-level language modeling task derived from The Complete Works of William Shakespeare as in (Caldas et al., 2018). It is structured for next-character prediction and is commonly used to evaluate federated learning methods in natural language processing tasks.The dataset is partitioned between 1,129 users.The dataset contains a total of 4,226,15 samples with 3,743.2 samples per user.

**Experimental Analysis of Line Search Steps in the SLS Optimizer**

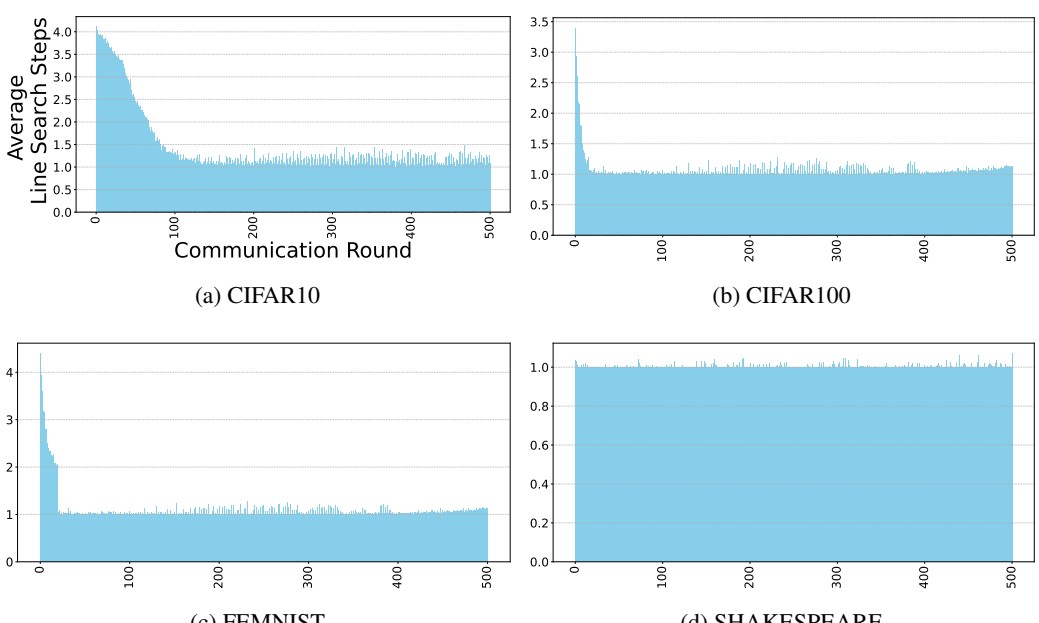

(a) CIFAR10          (b) CIFAR100

(c) FEMNIST          (d) SHAKESPEARE

Figure 4: Average Line Search Steps vs Communication Rounds

- In Figure 4, we evaluate the average number of line search steps (retries) per gradient step update per client during training with the **FedExpSLS** algorithm. As shown, the behavior of the SLS optimizer varies across different datasets:

- **CIFAR10 :** We observed a higher number of line search steps during the initial rounds of training. After approximately 100 rounds, this count rapidly declines and stabilizes at around one line search step per gradient step update per client. The plot 4a shows that the optimizer tunes the learning rate during first 100 rounds of training.

- **CIFAR100 and FEMNIST :** With CIFAR100 and FEMNIST dataset, the number of line search steps drops sharply from around 4 to approximately 1 within the first 50 training rounds. The drop of retry count suggests faster convergence by the optimizer.

- **SHAKESPEARE :** The line search step count remains around 1 consistently throughout the training.

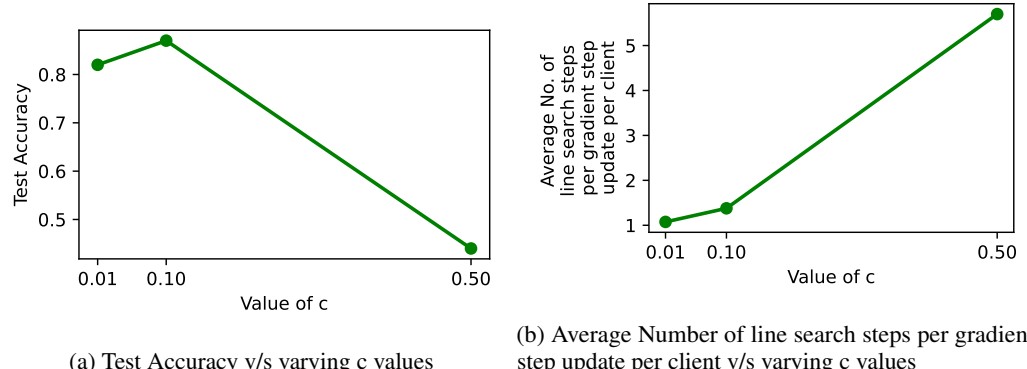

(a) Test Accuracy v/s varying c values

(b) Average Number of line search steps per gradient step update per client v/s varying c values

Figure 5: CIFAR-10 experiments with varying $c$ values

From figure 5a and 5b, we observe that increasing the value of the hyperparameter $c$ leads to a decline in test accuracy and an increase in the line search steps when using the **FedExpSLS** algorithm.

