# OpenReview forum: "Painless Federated Learning: An Interplay of Line-search and Extrapolation"
_ICLR.cc/2026/Conference — ICLR 2026 Conference Withdrawn Submission_

### Official Review · Reviewer_XMg7 · 2025-10-16

**Soundness:** 3
**Presentation:** 2
**Contribution:** 2
**Rating:** 2
**Confidence:** 4

**Summary:**

The paper introduced two algorithms FEDSLS and FEDEXPSLS, proved their convergence property respectively in theory and offer empirical evidence to support the claims. FEDSLS is based on the famous FedAvg algorithm with Armijo line search applied on a client level in a stochastic fashion. FEDEXPSLS on the other hand is based on FedExP, which combines line search and extrapolation as an FL algorithm.

**Strengths:**

(1) The authors proposed two novel algorithms, and provide corresponding analysis under a set of assumptions, the assumptions are clearly stated and the theorems are proved rigorously.

(2) The paper combines the line search with extrapolation and obtain state of the art results.

(3) Empirical results are there to further validate the theoretical claims.

**Weaknesses:**

(1) My primary concern is the set of assumptions used in the paper, the authors assume interpolation condition in all cases of their convergence guarantee. Although previous studies have provide certain justifications, such an assumption is not likely to be checked in general. In fact, I do not quite understand why would such an assumption is needed in the case of FedSLS and FedExPSLS. The authors mentioned FedExProx, and in that case this assumption is needed because of the algorithm connects to the parallel projection algorithm to solve the convex feasibility problem where interpolation holds. However, in this case the two algorithms are based on the FedAvg and FedExP, which does not require such an assumption. It this sense, it seems to me that the acceleration effect of line search is based on this condition, but not the trick itself.

(2) The author claimed that the empirical performance is improved, I suppose here the authors are referring to the iteration complexity, i.e., the proposed algorithms take less communication rounds to converge. This is not a fair comparison because the amount of work performed by each client in each round is different. For each client, we are replacing the original solver SGD with SGD-Armijo, which requires many additional forward passes to ensure that the criteria is met. This causes additional overhead, which could be huge. In my opinion, a comparison of the overall computational needed should be provided.

(3) As I have mentioned, the algorithm seems to be a direct consequence of replacing the local solvers from SGD to SGD with Armijo, which is already studied by existing literatures. The extension seems to be pretty straight forward, and the benefits are not properly justified , despite its reliance on a very restrictive assumption that may not be needed.

**Questions:**

(1) In theorem 4, I do not get why this is a convergence guarantee, it seems that the iteration K appears on the numerator if we fix $\eta_{l_\max}$ properly, and in this case the RHS becomes increasing in K. Could the author clarifies how should we interpret this theorem?

(2) Could the authors explain why the proposed algorithm requires interpolation condition to converge. What changed compared to FedExP or FedAvg?

(3) Could the authors show what the overall computational complexity of the algorithm is, and compare to other FL baselines? Is there a way to quantify the number of additional overhead needed for the Armijo subroutine?

---

> ### Author Response · Authors · 2025-11-16
> **Rebuttal: Addressing concerns on the necessity and (in-)sufficiency of interpolation condition**
>
> Thank you for your time to evaluate our submission and noting that the assumptions are clearly stated and the theorems are proved rigorously. We are happy to address your concerns.
>
> >>My primary concern... trick itself.
>
> Very humbly, the reviewer's reading that in FedExProx (Li et al. (2024)) "this assumption is needed because the algorithm connects to the parallel projection algorithm to solve **the convex feasibility problem (CFP) where interpolation holds**" is incorrect because the solution of CFP is only a common feasible solution, not necessarily a common minimizer that the interpolation offers. The interpolation condition used by FedExProx is a theoretical tool for deriving convergence, as seen in Remark 3 therein. Previously, FedExp (Jhunjhunwala et al., 2023) used it to derive the approximate projection condition (relation (6) there) and thereby the extrapolated learning rate. Both of them justify this by citing an overparameterized regime, which is typically the case when training deep models on FL clients. The interpolation was also used in the convergence proof of certain centralized methods, such as those presented by Vaswani et al. (2020), where no CFP was involved.
>
> We argue that the reviewer's interpretation, which suggests that the acceleration obtained by FedSLS/FedExpSLS is based on the interpolation condition alone rather than the line search, is incorrect.
>
> + First, see FedAvg proof in Karimireddy et al., 2020. Therein, the bounded heterogeneity assumption for convex and $\beta$-smooth functions in Appendix D.1 is given as:
>     $$\frac{1}{N}\sum_{i=1}^N\|\|\nabla f_i(x)\|\|^2 \le \frac{2}{N}\sum_{i=1}^N \|\|\nabla f_i(x^\*)\|\|^2 + \frac{2}{N}\sum_{i=1}^N \|\|\nabla f_i(x) - \nabla f_i(x^\*)\|\|^2 {\le}\frac{2}{N}\sum_{i=1}^N \|\|\nabla f_i(x^\*)\|\|^2+ 4\beta\bigl(f(x) - f^\*\bigr).$$
>    Under interpolation, $\frac{2}{N}\sum_{i=1}^N \|\|\nabla f_i(x^\*)\|\|^2=0$. This results in $G=0$ and $ B^2\geq 2$ in relation (14). Substituting $G=0$ there, when $f_i$ are convex, the convergence bound in Theorem V (Appendix D.2.) for $\eta_l=\frac{1}{(1+B)^2 8\beta K\eta_g}$ and communication rounds $T\ge 1$ reduces to
>     $$E\left[f(\bar{x}^T)\right] - f(x^\*) \le \mathcal{O}\left(\frac{MD}{\sqrt{TKS}}+\frac{B^2 \beta D^2}{T} \right).$$
>     Here $M:=\sigma^2(1+\frac{S}{\eta\_g^2})$ and $D:=\|\|x_0-x^\ast\|\|^2$. Clearly, this rate is not deterministic even with interpolation due to the presence of $M$, which depends on the gradient noise $\sigma^2$.
>
> + Now, we argue that the deterministic rate in FedExp is due to its full gradients, and interpolation alone is not sufficient. See Theorem 1 therein. With **full- gradients and full participation**, convex $f_i$s, under $L$-smoothness, bounded data heterogeneity at the optimum, and $\eta_\ell \le \frac{1}{6\tau L}$, the iterates ${\mathbf{w}^{(t)}}$ satisfy:
>     $$f(\bar{\mathbf{w}}^{(T)}) - f^* \le \mathcal{O}\left(\frac{||\mathbf{w}^{(0)} - \mathbf{w}^*||^2}{\eta_\ell \tau \sum_{t=0}^{T-1}\eta_g^{(t)}}\right)+\mathcal{O}\left(\eta_\ell^{2}\tau(\tau - 1)L{\sigma_\ast}^2\right)+ \mathcal{O}\left(\eta_\ell\tau{\sigma_\ast}^2\right)$$
>     where $\tau$ is the number of local updates, $\eta_g^{(t)}$ is the server step size at round (t), and $\bar{\mathbf{w}}^{(T)}=\frac{\sum_{t=0}^{T-1} \eta_g^{(t)} \mathbf{w}^{(t)}}{\sum_{t=0}^{T-1} \eta_g^{(t)}}$. Under interpolation, $\sigma_\ast=0$, resulting in a deterministic rate.
>
>     Now, had there been stochastic gradient updates on the clients, the error ball would have an additional $\mathcal{O}(\sigma_l^2)$ term, free from ${\sigma_\ast}^2$, where $\sigma_l^2$ is the clients' gradient noise. Clearly, $\sigma_\ast=0$ would not remove this term to obtain a deterministic rate. Kindly refer to the cited papers in case of any notational doubt.
>
> + We also discussed in Section 4.2 that Li et al. (2024) are able to obtain linear convergence because they assume the proximal problem at all clients is **solved exactly**, i.e., by controlling the noise of client local solvers.
>
> + Finally, in our case, if we removed line search, then FedSLS is analogous to FedAvg, which, as shown above, can not have a deterministic rate by interpolation alone. Similarly, because we have partial client participation, we do need a line search to tame the gradient noise. Essentially, we needed interpolation to translate the objective gap from $f(w_t)-\frac{1}{S}\sum_{i\in\mathcal{S}_t}f_i(w\_{t, K}^i)$ to $f(w_t)-f(w^*)$ to show the effect of using Armijo line-search with SGD on clients.
>
> References:
> (Jhunjhunwala et al., 2023) Fedexp: Speeding up federated averaging via extrapolation. ICLR, 2023.
>
> (Karimireddy et al., 2020) Scaffold: Stochastic controlled averaging for federated learning. ICML, pages 5132–5143, PMLR 2020.
>
> (Li et al., 2024) The power of extrapolation in federated learning. NeurIPS 2024.
>
> (Vaswani et al., 2020) Painless stochastic gradient: Interpolation, line-search, and convergence rates. NeurIPS, 2019.

---

> > ### Author Response · Authors · 2025-11-16
> > **Continuation from the above rebuttal.**
> >
> > >>Computational overhead
> >
> > In Appendix F, we have included the average number of forward passes for line search (retries) per gradient update per client for FedExpSLS. We observed a marginally higher number of retries during the initial rounds of training for the vision datasets, whereas almost negligible overhead for training the RNN on Shakespeare. This count typically rapidly declines and stabilizes at around one line search per gradient update per client. For CIFAR-100 \& FEMNIST, the line search count drops from 4 to 1 after the first 50 training rounds, and for the Shakespeare dataset, it remains 1 throughout. This ablation study sufficiently demonstrates that the marginal computational overhead up to training stabilization is a tolerable overhead for achieving better time-to-accuracy/convergence. Notwithstanding, our method is also able to offer a superior theoretical guarantee thanks to the marginal *initial* overhead.
> >
> > >> As I have mentioned...needed.
> >
> > Firstly, we have amply clarified the necessity and sufficiency of the assumptions above.
> > Now, in FL, the clients only have access to noisy estimates of their local objectives $f_i(w,\xi)$ due to data sampling, making standard line search criteria, which depend on full function evaluations $f_i(w)$ or descent directions, harder to satisfy and analyze. Furthermore, each FL client $i=1,\dots,N$ has a distinct data distribution and therefore a unique local objective function $f_i$. As a result, the step size chosen by local Armijo conditions may not be aligned with descent directions for the global objective. This issue of misalignment makes the analysis of convergence particularly intricate. Essentially, our proof of linear convergence carefully accounts for the local gradient noise and cross-client statistical heterogeneity using the stochastic line search and interpolation, respectively. We humbly submit that the extension may appear practically straightforward, but it is not too straightforward to analyse theoretically.

---

> > > ### Author Response · Authors · 2025-11-17
> > > **Response to the questions raised**
> > >
> > > >> (1) In Theorem 4, I do not get ... how should we interpret this theorem?
> > >
> > > We guess this question may have arisen due to a notational misunderstanding by the reviewer. The concern stems from the appearance of the number of local steps $K$ in the numerator of the convergence bound, which suggests that the bound becomes worse as $K$ increases. This aligns with the observation that, during training of FL models, when the number of local steps is large, convergence deteriorates due to larger client drift before a synchronization. This leads to misalignment between the client-side minimization problem and the global minimization problem. This is in line with the convergence bounds for FedAvg (Karimireddy et al., 2020) and FedExp (Jhunjhunwala et al., 2023), which also worsen with increasing local number of steps.
> > >
> > > >> (2) Could the authors explain why the proposed algorithm requires an interpolation condition to converge? What changed compared to FedExP or FedAvg?
> > >
> > > The empirical convergence is independent of the interpolation condition. For theoretical guarantees, as amply discussed above, it is required to bound the objective gap. We would refer the reviewer to lines 290-297 in Section 4.2 of our paper. Reiterating, the interpolation assumption is not a convergence enabler, but merely a tool to bound the heterogeneity in place of an explicit $G-B-$ bounds, as employed by other papers such as FedProx, FedExp, etc., which are fine with using the explicit bound, as they do not aim for deterministic rates with partial participation. The main theoretical change is that we have bounded heterogeneity through line search and interpolation together, rather than explicit $G-B-$ bounds, resulting in our deterministic convergence rates. The algorithmic difference from those works is sufficiently highlighted.
> > >
> > > >>(3) Could the authors show what the overall computational complexity of the algorithm is, and compare to other FL baselines? Is there a way to quantify the number of additional overhead needed for the Armijo subroutine?
> > >
> > > In experiments, we use Algorithm 2 to implement SGD-Armijo, where the search is over the grid $\{\eta_{l_{\max}}, \beta \eta_{l_{\max}}, \beta^2\eta_{l_{\max}},...\}$. Thus, we may track the computational complexity by tracking the operations that are performed at the start of each local step on each client (in Algorithm 2):
> > >
> > > * Line 5: compute $ \nabla f_i(w_{k-1}^i, b_k) $ → **1 gradient evaluation**.
> > > * Line 6: compute $ f_i(w_{k-1}^i, b_k) $ for RHS of Armijo → **1 function evaluation**
> > >     These are computed **once** per local step regardless of backtracking.
> > >
> > > * Each backtracking iteration tries a new $\eta$ to compute $w_k^i = w_{k-1}^i - \eta \nabla f_i(w_{k-1}^i, b_k)$. For that, it evaluates $ f_i(w_k^i, b_k) $ to check the Armijo condition. Thus, **1 function evaluation per trial**.
> > >
> > > Let’s define $T_k$ as the number of backtracking iterations at step $k$ (depends on how many times the loop runs before satisfying the Armijo condition). Then:
> > >
> > > * **Total function evaluations per step** = $( 1 + T_k )$.
> > > * **Total gradient evaluations per step** = 1 (shared across all backtracking attempts)
> > >
> > > ## Total Cost per Client per Local Round
> > >
> > > Let:
> > >
> > > * $K$: number of local steps per client per communication round.
> > > * $\bar{T}$: average number of backtracking iterations per step (i.e., $ \bar{T} = \frac{1}{K} \sum_{k=1}^K T_k )$
> > >
> > > Then, for **one client** in one communication round, there are $K\cdot (1 + \bar{T})$ function evaluations and $K$ gradient evaluations.
> > >
> > > ## Total Cost per Client (per round):
> > >
> > > $$\text{Total cost} = {K} + K(1 + \bar{T}) = K (2 + \bar{T})$$
> > >
> > > Whereas, for FedAvg, the cost per client per round is $K$.
> > >
> > > As our experiments in Appendix F show, $\bar{T}$ reduces with training rounds as the number of backtracking loops reduces after the initial phase of training. This additional computation is what enables FedSLS/FedExpsSLS to outperform their original counterparts.

---

> ### Author Response · Authors · 2025-11-17
> **Thank you!**
>
> We sincerely thank the reviewer again for reading our submission and raising the questions. We are glad to have answered the pertinent questions, and we hope that this will also clarify similar doubts that readers of our work on OpenReview may have. We will also include this entire discussion in an appendix in the manuscript.
>
> We hope that we have satisfactorily addressed the reviewer's concerns. We will be more than happy to promptly respond further should the reviewer have further questions.
>
> If you have no further concerns, we kindly request that you review the score.

---

### Official Review · Reviewer_PAVk · 2025-11-01

**Soundness:** 4
**Presentation:** 3
**Contribution:** 3
**Rating:** 4
**Confidence:** 4

**Summary:**

The paper proposes FedSLS, which is like FedAvg except the clients perform Armijo line search to set the learning rate. Aside from the practical benefit of not having to set a learning rate, the Armijo test helps the convergence proof by allowing the analysis to control the client drift rather than assuming bounded heterogeneity which is typically done in practice. This allows the authors to achieve linear convergence in the strongly convex setting even with partial client participation. FedSLS in some sense follows the prior work on stochastic line search and applies it to the federated setting, where inter-client variance (heterogeneity) is somewhat like the variance in the centralized setting. In addition to the theoretical guarantees, the authors are able to show the method performs well on experiments, in particular FedExpSLS.

**Strengths:**

- The algorithm makes sense and appears simple to implement, and shows that line search is a natural thing to try in FL
- Linear convergence under partial participation is somewhat expected given the analogous result in the centralized setting, but it is good to see the authors are able to derive it.
- FedExpSLS looks pretty strong, with solid convergence results compared to other methods
- The line search overhead seems to not be too much, again showing the algorithm is not too costly

**Weaknesses:**

- The work assumes that there is an optimum that is shared across all the clients. This is a pretty strong assumption
- Communication round-based plots are good, but there should probably be wall-clock based ones (or total client iteration-based) because of the additional client compute used by the line search
- Tuning grids should be included in the experiments
- Ideally in optimization, training runs should be compared against prior work in a leaderboard-like manner. See for example https://kellerjordan.github.io/posts/speedrun/ in the centralized setting. Is there something like this for the FL optimization literature? Without it, it is hard to trust results are significant
- In particular, the paper mentions that FedAdam underperforms, but for the dataset/tasks considered, it usually performs quite well, see

Reddi, Sashank, et al. "Adaptive federated optimization." arXiv preprint arXiv:2003.00295 (2020).

- I wonder how pretraining or LLM assistance would affect the empirical results? Both have become relatively standard in FL deployments. See

Nguyen, John, et al. "Where to begin? on the impact of pre-training and initialization in federated learning." arXiv preprint arXiv:2206.15387 (2022).

Hou, Charlie, et al. "Private federated learning using preference-optimized synthetic data." arXiv preprint arXiv:2504.16438 (2025).

Wu, Shanshan, et al. "Prompt public large language models to synthesize data for private on-device applications." arXiv preprint arXiv:2404.04360 (2024).

**Questions:**

See weaknesses

---

> ### Author Response · Authors · 2025-11-15
> **Rebuttal: Clarification on the interpolation condition and convergence with wall-clock time.**
>
> We thank the reviewer for the time taken to read our submission and for the encouraging remarks that the presented FedExpSLS algorithm looks pretty strong, with solid convergence results compared to other methods. Here we address the concerns raised by the reviewer.
>
> >>The work assumes that there is an optimum that is shared across all the clients. This is a pretty strong assumption.
>
> This is the **interpolation** condition. Although it appears theoretically strong, as noted by Arora et al. (2019) and Montanari and Zhong (2020), it is satisfied in the overparameterized regime, which standard deep models typically fit.
>
> Let us review the literature on the convergence theory of federated learning. There are mainly two tools to tackle the statistical heterogeneity-induced term in the error ball: a) G-B boundedness on the gradient norms across the clients, and b) the interpolation. In some sense, both these conditions are strong; however, we concur with the reviewer that the interpolation appears stronger. Recent works on FL invariably assume an interpolation condition, as seen in FedExp by Jhunjhunwala et al. (2023), FedExpProx by Li et al. (2024, 2025), and others. We needed this assumption to translate the objective gap $f(w\_t)-\frac{1}{S}\sum_{i\in\mathcal{S}\_t} f\_i (w\_{t,K}^i)$ to $f(w_t)-f(w^*)$, which is in line with the published works of Li et al. (2024, 2025). Wherever it is used in the literature, the standard justification is the overparameterized regime on clients with limited data in a federated setup.
>
> References:
>
> (Arora et al., 2019)  S. Arora, S. Du, W. Hu, Z. Li, and R. Wang. Fine-grained analysis of optimization and generalization for overparameterized two-layer neural networks. ICML, 2019.
>
> (Montanari and Zhong, 2020) A. Montanari and Y. Zhong. The interpolation phase transition in neural networks: Memorization and generalization under lazy training. The Annals of Statistics, 50(5):2816–2847, 2022.
>
> >>Wall-clock-based
>
> We would like to emphasize that taming statistical heterogeneity and achieving linear convergence with partial client participation certainly outweighs paying a small additional overhead in a federated setting. Unlike distributed machine learning, throughput has perhaps not been a core concern for an FL algorithm designer, as evidenced by almost every published work, where, to our knowledge, convergence in wall-clock time has never been a metric for evaluation. That said, as rightly noted by the reviewer, the additional cost of line search is quite minimal, which we have included in Appendix F.
>
> >>Tuning grids should be included in the experiments
>
> For each of our algorithms, we applied cosine annealing on the clients running SGD between every synchronization round. On the server, for the FedSLS algorithm, a simple tuning grid was used for the constant learning rate (LR), where a factor of 10 was used to move to the next grid point. For the FedExpSLS algorithm, the server LR is computed by the extrapolation rule. For the hyperparameter $c$ used by the Armijo line search algorithm, we have provided the tuning trade-off in Figure 5 in Appendix F. We thank the reviewer for suggesting this and will include it in the final version.
>
> >> Leaderboard-style evaluations
>
> Thank you for the suggestion and the reference to leaderboard-style evaluations in centralized optimization. We agree that standardized leaderboards can be valuable for benchmarking methods in a reproducible and transparent manner. However, in FL, such standardization is still evolving. To our knowledge, there is currently no widely accepted leaderboard for federated optimization. We evaluated our methods on widely used benchmarks under a consistent setup. A pointer to the code is included in the manuscript. We would like to acknowledge that your comment has motivated us to develop a leaderboard.
>
> >> In particular, the paper mentions that FedAdam underperforms, ..
>
> Please see our response to reviewer 8tcT.
>
> >>I wonder how pretraining or LLM assistance would affect ..
>
> First, let us note that our work primarily aims to provide a method for mitigating the slowdown caused by statistical heterogeneity, which is particularly relevant in the cross-device setting, such as that realized by telecommunication networks. In this setting, out of hundreds of clients, only a couple of tens are available in any synchronization round. Having said that, we acknowledge the reviewer's suggestion that starting the training from a pre-trained weight, or, for that matter, using the LLM-generated synthetic data, would be an interesting experiment. However, we are unsure if such additional experiments would have provided any significantly new insights to validate the proposed theory that our included experiments may have possibly missed.
>
> We will be happy to address any further concerns. If you have no other concerns, we kindly request that you review your score.

---

> > ### Comment · Reviewer_PAVk · 2025-11-21
> >
> > - On the FedAdam point, you mention
> >
> > > On the other hand, we have run only 500 rounds, after which the accuracy curve's slope remains positive, indicating that higher accuracy can be achieved with additional rounds.
> >
> > Why would you evaluate on a different round budget? It's better to emulate the experimental settings of prior work to ensure they are comparable. This is the exact kind of problem that leaderboard-style evaluations are meant to tackle. If there is no leaderboard then you should at least follow the settings of prior work. There is generally a large body of optimization papers which basically don't have any gain and as a community we should hold new proposed algorithms to a common standard.
> >
> > - On pretraining/synthetic data to mitigate heterogeneity, you should at least have a discussion/justification in your paper why you do not address it, since it is the most common tool to deal with heterogeneity today. Best to have an actual empirical comparison to make your justification stronger. They also consider the cross-device setting, so there is no setting mismatch here.

---

> > > ### Author Response · Authors · 2025-11-29
> > > **Response to the Official Comment**
> > >
> > > Dear reviwer PAVk,
> > >
> > > Thank you for taking the time to read our rebuttal. We appreciate your further comments. We respond to it below:
> > >
> > > > Why would you evaluate on a different round budget? It's better to emulate the experimental settings of prior work to ensure they are comparable. This is the exact kind of problem that leaderboard-style evaluations are meant to tackle. If there is no leaderboard, then you should at least follow the settings of prior work. There is generally a large body of optimization papers that don't yield significant gains, and as a community, we should hold new proposed algorithms to a common standard.
> > >
> > > We again appreciate your encouragement for a leaderboard-style evaluation. Indeed, when there is no such leaderboard, we should follow the settings of the prior work in a fair and objective manner. Now, let's consider this objectivity here
> > > + Firstly, as we mentioned on lines  470 and 471 in our paper, we included FedAdam for the CIFAR100 and Shakespeare datasets to compete in the settings of higher heterogeneity and NLP tasks with the motivation that Adam is known to outperform SGD with momentum for NLP tasks (Kunstner et al., 2024), and, as the FedAdam paper claimed, they designed the algorithm with the motivation to tackle heterogeneity. We did not include FedAdam in other cases, where it would have a potential disadvantage even to begin with, such as the CIFAR-10 and FEMNIST datasets.
> > > + Let us refer to Table 1 of the FedEXP paper, which compares different methods in terms of time to Target accuracy, which is quite desirable, as one may not need to train for 4000 rounds to achieve some accuracy by way of extreme hyperparameter tuning to the final bit.
> > > + This is exactly the standard: to show the efficacy of a new optimization algorithm, we compare it against the existing methods in terms of training performance to reach a target accuracy as visualized in those accuracy curves. We have adopted this standard absolutely.
> > >
> > > Having said that, here are our results for training ResNet18 using FedSLS, FedExpSLS, and FedAdam for 4000 rounds on CIFAR100 data, while maintaining the standard hyperparameter selections. The table below shows the maximum validation accuracy, which is again a standard benchmark metric, achieved over the entire training trajectory:
> > > | FedAdam | FedSLS | FedExpSLS |
> > > |---------|--------|-----------|
> > > | 53.1    | 51.7   | 55.2   |
> > >
> > > Furthermore, our results for training the employed LSTM for 1200 rounds on Shakespeare data using the same methods are  below:
> > > | FedAdam | FedSLS | FedExpSLS |
> > > |---------|--------|-----------|
> > > | 56.8    | 56.9   | 57.6   |
> > >
> > > As anticipated, our methods outperform FedAdam over an extended optimization trajectory, too.
> > >
> > > Kunstner, Frederik, et al. "Heavy-tailed class imbalance and why Adam outperforms gradient descent on language models." Advances in NeurIPS 2024.
> > >
> > > >On pretraining/synthetic data to mitigate heterogeneity, you should at least have a discussion/justification in your paper why you do not address it, since it is the most common tool to deal with heterogeneity today. Best to have an actual empirical comparison to make your justification stronger. They also consider the cross-device setting, so there is no setting mismatch here.
> > >
> > > Very humbly and respectfully, we would contend the argument that pretraining/synthetic data is the "most common tool" to deal with heterogeneity today. Let us discuss them here,
> > > Firstly, regarding the synthetic data, the primary motivation in the papers you referred to — Hou et al. and Wu et al. — is to present a use case of LLMs for generating local data for differential privacy. This is an LLM application, and neither it aims nor it discusses mitigating heterogeneity. (Unless we missed it, they do not even have the term "heterogeneity" used even once.)
> > > + Now, about pretraining the models, Nguyen et al. present their observations that if FL starts from a pretrained model, the effect of heterogeneity on the later part of training is lessened. Although the paper presented this observation without any theoretical discussion, our discussion provides an opportunity to explore why that might be the case. Notice that, when we have a centralized training for $k$ rounds out of $n$ total rounds, the round-wise convergence would have heterogeneity error for only $n-k$ rounds. Clearly, as $k\rightarrow n$, the heterogeneity error would tend to $0$. Isn't this unsurprising? Can we safely say that the papers such as FedExp, FedExprox, etc., which appeared later and offer principled approaches to tackling heterogeneity, did not consider this work primarily because it was an unsurprising observation? For FedSLS and FedExpProx, our primary motivation was to achieve a deterministic convergence rate that mitigates the heterogeneity, as captured in theory, by subsuming the heterogeneity error over the entire optimization dynamics.
> > >
> > > We hope that we have addressed your concerns. If no further concerns, kindly review your score.

---

### Official Review · Reviewer_8tcT · 2025-11-05

**Soundness:** 2
**Presentation:** 2
**Contribution:** 2
**Rating:** 4
**Confidence:** 4

**Summary:**

The paper introduces two new algorithms, Federated Stochastic Line Search (FEDSLS) and Federated Extrapolated Stochastic Line Search (FEDEXPSLS), that adapt stochastic Armijo line search for federated optimization. By applying stochastic line search locally at clients, FEDSLS dynamically adjusts learning rates, mitigating both gradient noise and data heterogeneity. FEDEXPSLS extends this with extrapolation of the server learning rate to further accelerate convergence. Theoretically, both methods achieve deterministic convergence rates even under partial client participation. Experiments on CIFAR-10, CIFAR-100, FEMNIST, and Shakespeare benchmarks demonstrate that FEDEXPSLS consistently outperforms existing methods like FedAvg, FedExp, and FedProx in both training loss and test accuracy, with negligible computational overhead.

**Strengths:**

* Introduces the Armijo condition, which can be used to overcome the bias in SGD without the sample-wise interpolation for a single local solver.
* Theoretical guarantees showing that FedSLS and FedExpSLS achieve deterministic rates in FL under standard assumptions along with the Armijo condition and the interpolation condition.
* Proposed methods are evaluated on multiple datasets (CIFAR-10/100, FEMNIST, Shakespeare) and models (ResNet-18, LSTM, etc.), consistently outperforming FedAvg, FedExp, FedProx, and FedAdam in both training loss and test accuracy,

**Weaknesses:**

**Lack of clear motivation and positioning:**
The motivation for the work feels underdeveloped. The introduction reads more like a problem formulation than a compelling argument for why this specific direction is needed. Many algorithms already aim to speed up federated learning, so it’s unclear why line search, and particularly Armijo line search, is the right tool for the job. The paper would benefit from a clearer explanation of what gap this approach fills and why combining line search with extrapolation is conceptually or practically appealing compared to existing methods.

**Limited discussion around the Armijo condition:**
Definition 3 is theoretically interesting, but it’s not clear when or how this condition holds in practice. For example, do deep neural networks approximately satisfy it? If not, what types of models or loss functions make this assumption reasonable? Some empirical evidence, perhaps showing the frequency of Armijo condition satisfaction during training, would help clarify the practical relevance of this assumption.

**Restricted experimental scope:**
The experiments are limited to small-scale benchmarks like CIFAR, FEMNIST, and Shakespeare. While these are common in FL papers, they don’t provide much insight into performance or scalability in more realistic cross-device settings. Including larger or more modern datasets such as StackOverflow, Reddit, or Google Landmark-v2 would make the results more convincing. Also, the surprisingly poor performance of FedAdam raises some concerns about hyperparameter tuning or implementation consistency.

**Questions:**

N/A

---

> ### Author Response · Authors · 2025-11-15
> **Rebuttal: Clarifications on motivations and positioning, and clarifications on the experimental results.**
>
> We thank the reviewer for the time taken to read our submission. Below, we address the concerns that have been raised.
>
> >>**Lack of clear motivation and positioning**: The motivation ...
>
> From our introduction section, see lines 75 to 78, among the many existing algorithms proposed to accelerate federated learning, a major gap is that all but one do not offer deterministic rates of convergence for federated iterates. It raises a natural question -- can we investigate adapting a tool from the centralized setting to the federated setting to fill this gap? Recall that several developments in federated optimization were made possible by this core design principle: adapting a centralized algorithm to a federated one.
>
> Our algorithm is only the second to achieve linear convergence while tolerating partial client participation in a novel manner. The core mechanism enabling this is the Armijo line search technique.
>
> >>**Limited discussion around the Armijo condition**: Definition (3)
>
> It appears that the reviewer has a misunderstanding of Definition (3). This definition **is not assumed to be a property of the model, neural network, or loss function, etc.** It is **ensured by the algorithm**, which implements stochastic adaptation of the classical Armijo condition (Armijo, 1966); see lines 3 to 6 in Algorithm 2 in Appendix A. More emphatically,
>
> + **Definition 3 is operational**, not assumed. It is **actively checked and enforced** by each client during training.
>
> + Algorithms 4 and 5 (FedSLS and FedExpSLS) explicitly call Algorithm 2, which performs a line search to find a step size satisfying Definition 3 on the stochastic loss.
>
> + This means that Definition 3 becomes a **criterion for acceptance** of a particular step size, not a property that must hold a priori.
>
> This is similar to (Paquette and Scheinberg, 2020; Vaswani et al., 2020; Jin et al., 2021):
>
> (Paquette and Scheinberg, 2020) Courtney Paquette and Katya Scheinberg. A stochastic line search method with expected complexity analysis. SIAM Journal on Optimization, 30(1):349–376, 2020.
>
> (Vaswani et al., 2020) Sharan Vaswani et al. Painless stochastic gradient: Interpolation, line-search, and convergence rates. NeurIPS, 2019.
>
> (Jin et al., 2021) Billy Jin et al. High probability complexity bounds for line search based on stochastic oracles. NeurIPS 2021.
>
> (Armijo, 1966) Larry Armijo. Minimization of functions having lipschitz continuous first partial derivatives. Pacific Journal of
> mathematics, 16(1):1–3, 1966.
>
>
> >>**Restricted experimental scope**
>
> Again, it appears that the reviewer is not clear about this query. The cross-device setting is precisely the one that we have employed: 100 clients with data distributed over each of them using a Dirichlet-$\alpha=0.3$, representing more than a moderate heterogeneity (on average, 3 classes out of 10 would be available on a client, see (Caldas, 2018)), followed by sampling 20 clients, see lines 409 to 416. We have used the standard representative datasets for both computer vision and NLP tasks. See the representative literature that covers the developments over the last 5 years:
>
> * The **SCAFFOLD** paper primarily reports results on **EMNIST**.
> * **FedExp** evaluates on **EMNIST, CIFAR-10, CIFAR-100, CINIC-10**.
> * **FedExpProx** is tested only on **synthetic or overparameterized linear models**.
>
> We agree that evaluating on other datasets suggested by the reviewer would provide additional insights; however, our focus is on theoretical contributions for the line-search-based FL algorithm, and we believe these experiments sufficiently support our findings.
>
> (Caldas, 2018) Sebastian Caldas et al. Leaf: A benchmark for federated settings. 2018.
>
> **On FedAdam Performance**:
>
> We acknowledge the reviewer’s observations on the accuracy mismatch for FedAdam in some of our experiments and those reported elsewhere. However, this is not really a mismatch. To clarify, we have used the same models: ResNet-18 and the same RNN used in the FedAdam paper for CIFAR-100 and Shakespeare datasets, respectively. We employed the same tuning strategy and conducted the same grid search across the baselines, ensuring a fair comparison. However, the FedAdam paper reports validation accuracy over 4,000 rounds for CIFAR-100 and 1,200 rounds for Shakespeare, with LRs tuned for optimal performance in the final 100 rounds. On the other hand, we have run only 500 rounds, after which the accuracy curve's slope remains positive, indicating that higher accuracy can be achieved with additional rounds. Furthermore, we note that the results reported in the FedAdam paper do not mention the statistical significance of their findings, whereas we run our experiments 5 times with different random seeds and report the average and standard deviation. We will conduct additional experiments and report the results for a matching number of rounds in the final version.
>
> If you have no further concerns, we kindly request that you review the score.

---

### Author Response · Authors · 2025-12-04
**Summary**

Dear AC & Reviewers,

We appreciate your feedback on our submission. We briefly summarize our contributions and the discussion regarding the concerns raised.

**Our Contributions:**

(1) We proposed the FedSLS and FedExpSLS algorithms that use stochastic Armijo line-search as a local solver.

(2) We show that Armijo line search subsumes the noise due to stochastic gradients and achieves deterministic convergence rates in expectation without the sample-wise interpolation condition.

(3) Under the client-wise interpolation assumption, FedSLS and FedExpSLS achieve linear convergence rates for strongly convex objectives and deterministic GD rates for convex and non-convex objectives.

(4) Empirically, the FedExpSLS algorithm consistently outperforms the state-of-the-art algorithms.

---

Reviewer **8tcT** notes the deterministic convergence guarantees of our algorithms under partial participation, and the consistent empirical gains across benchmarks. The reviewer’s main concerns involved the clarity of motivation, interpretation of the Armijo condition, scope of experiments, and performance of FedAdam. We clarified that our motivation directly targets a key remaining gap in FL- **achieving deterministic linear convergence**- and that Armijo line search is a natural tool to adapt from centralized optimization toward this end. We clarified that Definition 3 is **not an assumption** on the model or loss, but rather the Armijo condition **enforced by the algorithm** through stochastic function and gradient computation. Regarding the experimental scope, we emphasized that our setup corresponds directly to the **cross-device FL regime** widely used in the literature, and we follow representative prior works that use the same datasets and settings.

---

Reviewer **PAVk** acknowledges the strong empirical performance of FedSLS/FedExpSLS and the minimal line-search overhead. The reviewer’s main concerns—(i) the interpolation assumption, (ii) wall-clock or total client iteration-based plots, (iii) tuning grid, (iv) FedAdam comparison, and (v) pretraining/synthetic data—have been fully addressed in our responses. We clarified that **interpolation is now standard in FL theory** and we require this assumption for translating client-side objective gap to the global objective gap. Appendix F includes a discussion on the negligible cost of line search. We included FedAdam evaluations under the same round budgets and configurations as prior work, and show that FedExpSLS and FedSLS outperform FedAdam over the full trajectory on CIFAR-100 and Shakespeare. We do not consider pretraining/LLM-generated data, as these approaches are orthogonal to our contribution and not primarily aimed at mitigating heterogeneity.

---

Reviewer **XMg7** remarks that "*the assumptions are clearly stated, and the theorems are proved rigorously.*" The reviewer expresses concern about why we assumed the interpolation condition and suggested that the interpolation assumption holds for the convex feasibility problem, and "*it seems to me that the acceleration effect of line search is based on this (interpolation) condition, but not the trick itself.*" We have rigorously clarified that this assumption holds for deep neural networks under an overparameterized regime and justified that both FedExp and FedAvg, under a stochastic setting, do not achieve deterministic convergence guarantees solely under the interpolation assumption. We also explained why this assumption is necessary for our analysis. The reviewer raised concerns regarding the fairness of the comparison with the state of the art owing to the overhead of the line search. We point out that the overhead is minimal and **the per-round client cost of our method is only a factor of $(2+\bar{T})$ higher than FedAvg**, i.e., $K(2+\bar{T})$ vs. $K$, where $\bar{T}$ is the average number of Armijo line-search calls. The reviewer noted that **the iteration K appears in the numerator** in the convergence bounds. We again note that this is in alignment with the convergence bounds for FedAvg (Karimireddy et al., 2020) and FedExp (Jhunjhunwala et al., 2023), which also worsen with increasing local number of steps. This occurs because, during the training of FL models, when the number of local steps is large, convergence deteriorates due to larger client drift before synchronization.

----

We hope that we have satisfactorily addressed all your concerns. We are thankful for your participation in the discussion.

---

### Note · Authors · 2026-01-26

**Comment:**

The reviews did not reflect the paper's merit.

**Withdrawal Confirmation:**

I have read and agree with the venue's withdrawal policy on behalf of myself and my co-authors.

---

### Meta-Review · Area_Chair_B7gD · 2026-01-06

**Summary:**

This paper proposed a class of federated stochastic line search (i.e., FEDSLS and FEDEXPSLS) methods based on the Armijo condition. It provided the convergence analysis for the proposed FEDSLS and FEDEXPSLS methods. It also provided some numerical experiments to demonstrate efficiency of the proposed algorithms.

Overall, this paper uses the Armijo line search to FedAvg algorithm, which is trite.  In the convergence analysis, some conditions such as Assumption 6 in the paper are strict. In the experiment, some comparisons such as Scaffold [1] are missing.

[1] Scaffold: Stochastic controlled averaging for federated learning, ICML 2020

**Reviewer Concerns:**

Although the authors have provided rebuttals to address some reviewers' concerns, there exist some concerns still are not addressed. For example, Reviewer PAVk believe that  there is an optimum that is shared across all the clients, which is a pretty strong assumption.

**Reviewer Scores:**

Although the authors have provided rebuttals to address some reviewers' concerns, there exist some concerns still are not addressed. Thus, the likelihood of drastically changing the score is very low.

---

### Decision · Program_Chairs · 2026-01-26

Reject